# Adaptive Threshold Model in Google Earth Engine: A Case Study of *Ulva prolifera* Extraction in the South Yellow Sea, China

Guangzong Zhang [1], Mengquan Wu [2] , Juan Wei [3], Yufang He [1], Lifeng Niu [1], Hanyu Li [1] and Guochang Xu [1,*]

[1] Institute of Space Science and Applied Technology, Harbin Institute of Technology (Shenzhen), Shenzhen 518055, China; zhanggz1994@163.com (G.Z.); 19b958025@stu.hit.edu.cn (Y.H.); 20b958034@stu.hit.edu.cn (L.N.); 20s058100@stu.hit.edu.cn (H.L.)

[2] School of Resources and Environment Engineering, Ludong University, Yantai 264025, China; ld_wmq@ldu.edu.cn

[3] College of Marine Geosciences, Ocean University of China, Qingdao 266000, China; weijuan@stu.ouc.edu.cn

* Correspondence: xuguochang@hit.edu.cn; Tel.: +86-1838-910-1556

**Abstract:** An outbreak of *Ulva prolifera* poses a massive threat to coastal ecology in the Southern Yellow Sea, China (SYS). It is a necessity to extract its area and monitor its development accurately. At present, *Ulva prolifera* monitoring by remote sensing imagery is mostly based on a fixed threshold or artificial visual interpretation for threshold selection, which has large errors. In this paper, an adaptive threshold model based on Google Earth Engine (GEE) is proposed and applied to extract *U. prolifera* in the SYS. The model first applies the Floating Algae Index (FAI) or Normalized Difference Vegetation Index (NDVI) algorithm on the preprocessed remote sensing images and then uses the Canny Edge Filter and Otsu threshold segmentation algorithm to extract the threshold automatically. The model is applied to Landsat8/OLI and Sentinel-2/MSI images, and the confusion matrix and cross-sensor comparison are used to evaluate the accuracy and applicability of the model. The verification results show that the model extraction of *U. prolifera* based on the FAI algorithm has higher accuracy ($R^2 = 0.99$, RMSE = 5.64) and better robustness. However, when the average cloud cover is more than 70% in the image (based on the statistical results of multi-year cloud cover information), the model based on the NDVI algorithm has better applicability and can extract the algae distributed at the edge of the cloud. When the model uses the FAI algorithm, it is named FAI-COM (model based on FAI, the Canny Edge Filter, and Otsu thresholding). And when the model uses the NDVI algorithm, it is named NDVI-COM (model based on NDVI, the Canny Edge Filter, and Otsu thresholding). Therefore, the final extraction results are generated by supplementing NDVI-COM results on the basis of FAI-COM extraction results in this paper. The F1-score of *U. prolifera* extracted results is above 0.85. The spatiotemporal distribution of *U. prolifera* in the South Yellow Sea from 2016 to 2020 is obtained through the model calculation. Overall, the coverage area of *U. prolifera* shows a decreasing trend over the five years. It is found that the delay in recovery time of *Porphyra yezoensis* culture facilities in the Northern Jiangsu Shoal and the manual salvage and cleaning-up of *U. prolifera* in May are among the reasons for the smaller interannual scale of algae in 2017 and 2018.

**Keywords:** Southern Yellow Sea; *Ulva prolifera*; Otsu thresholding; Canny Edge Filter; floating algae index; normalized difference vegetation index; Google Earth Engine

## 1. Introduction

In recent years, green macroalgae blooms (MABs) caused by the green tide have been widely reported and have become major global marine disasters [1–3]. Green tide is a kind of harmful algae bloom, and *Ulva prolifera* is the dominant algal species involved in these blooms. Since 2007, *U. prolifera* has broken out in the South Yellow Sea of China for 13

consecutive years. The main characteristics of this marine disaster are rapid outbreak and wide distribution. Studies show that the disaster could quickly spread to most coastal cities on the Shandong Peninsula [4–6]. If *U. prolifera* is not treated in time, it will harm marine life, destroy aquaculture, block the river, and affect human life. The large-scale growth of *U. prolifera* could result in the surrounding seawater environment lacking oxygen and the production of allelochemicals that inhibit the reproduction of other phytoplankton algae and disrupt the coastal environment [7–9]. In addition, cleaning up *U. prolifera* poses a huge burden on the government [10,11]. There are about $3.6 \times 10^5$ tons of *U. prolifera* on the sea each year, and most decomposes into the environment, which has a negative impact on the ecology and economy of coastal cities [12–14].

In the past 10 years, many scholars have carried out various studies on the whole life cycle of *U. prolifera*. The results show that *U. prolifera* can be divided into five stages: "growth", "development", "outbreak", "decline", and "extinction" [15]. Remote sensing technology and field monitoring data showed that the earlier discovery *U. prolifera* was from the large-scale *Porphyra yezoensis* culture in Jiangsu shoal, and the *P. yezoensis* culture raft provided attachment conditions for the growth of *U. prolifera* [16]. The growth and drift of the early algae occur in April and May each year. In June, the growth rate of the *U. prolifera* increased day by day due to the abundant nutrients and suitable surface temperature, and it gathered in the South Yellow Sea at a large scale [17,18]. Then, in July and August, with the increase in the sea surface temperature and changes in environmental factors, such as a nutrient decrease, *U. prolifera* began to decay and die. A large amount of *U. prolifera* sank to the seabed and decomposed, and only a small portion accumulated the coast [19].

Remote sensing technology can quickly and dynamically monitor the growth cycle and coverage area of *U. prolifera*. The first step is to select the appropriate algorithm. The spectral characteristics of *U. prolifera* are very similar to those of green vegetation. Therefore, the Normalized Difference Vegetation Index (NDVI) algorithm and enhanced vegetation index (EVI) algorithm have been used in MERIS (Medium-Resolution Imaging Spectrometer), Aqua and Terra/MODIS (Moderate-Resolution Images Spectroradiometer), Landsat5/TM (Thematic Mapper), Landsat8/OLI (Operational Land Imager), and HJ-1/CCD (Charge-Coupled Device) satellite images to realize the monitoring of *U. prolifera*. [18,20–22]. After that, Hu et al. proposed a floating algae index (FAI) algorithm suitable for the MODIS satellite [23]. Compared with the traditional algorithms, this method has higher accuracy and is less sensitive to changes in environmental and observing conditions. Even in the presence of thin clouds, it can also detect algae. Affected by the environment, geography, and other factors, images in different periods with the same algorithms will have different values. Therefore, we need to set an optimal threshold. The same threshold may not be suitable for images on long-term sequences, so manual intervention is required to select the threshold. Qi et al. applied the FAI algorithm to MODIS images and used objective statistical methods to analyze the average coverage of *U. prolifera* in the South Yellow Sea from 2007 to 2015. The results showed that the coverage of *U. prolifera* was the largest in 2015 [24]. However, since 2015, questions have remained: whether the average coverage of *U. prolifera* will continue to increase, whether we can reduce the scale of *U. prolifera* outbreaks under the policy of early salvage and cleanup of *U. prolifera*, and whether one can quickly and dynamically grasp the scale information of *U. prolifera* after an outbreak.

Scholars have applied machine learning, deep learning, and cloud computing to the extraction and monitoring of *U. prolifera* [25,26]. Using the FAI algorithm, Qiu et al. established a machine learning model for the automatic continuous recognition of large algae with a multilayer perceptron, then applied this model to the Geostationary Ocean Color Imager (GOCI) satellite data. The results showed that the method has stronger robustness than the traditional threshold selection algorithm [27]. However, for machine learning methods based on training samples, the classification relies on a large number of training samples. However, training samples are usually laborious and expensive. Xu et al. applied the Otsu algorithm to a variety of satellite data to extract *U. prolifera* [28]. It was

found that this method can achieve dynamic threshold selection and extract *U. prolifera* more accurately. The Otsu algorithm overcomes the disadvantage of a uniform threshold in traditional algorithms but suffers from threshold anomalies when there is a large range of water pixels in the image.

To solve the above problems, an automated, accurate threshold calculation model needs to be proposed with a large amount of data for *U. prolifera* monitoring. The emergence of the Google Earth Engine (GEE) has changed the traditional remote sensing data processing mode, which is a cloud platform for huge geospatial data analysis [29,30]. This platform provides users with high-resolution satellite image data (Landsat series satellites, Sentinel series satellites) and performs image preprocessing operations on the cloud platform. By embedding the algae extraction algorithm, we can monitor the distribution of algae in a long-term sequence.

Therefore, this study is based on GEE and aims to (1) propose an adaptive threshold model and realize the automatic and rapid extraction of *U. prolifera* in the Southern Yellow Sea; (2) apply the model to high-resolution Landsat8/OLI and Sentinel-2/MSI images within GEE, and then evaluate the accuracy and applicability of this model for extracting *U. prolifera*, and (3) generate a distribution map of *U. prolifera* in the South Yellow Sea of China from 2016 to 2020, and analyze the spatial and temporal changes of *U. prolifera* over the five years.

## 2. Data and Methods

### 2.1. Study Area

This study area belongs to the Southern Yellow Sea (119–123° E, 32–37° N), a part of the Yellow Sea in China. The Southern Yellow Sea is a semi-enclosed shallow sea with an average depth of 44 m (Figure 1) [31,32], covering an area of 309,000 km². Influenced by the Yellow Sea Warm Current (YSWC), the Yellow Sea Cold Water Mass (YSCWM), the coastal current of the Yellow Sea, and the diluted current of the Yangtze River, the Southern Yellow Sea has complex hydrographic conditions [33]. In addition, the southern part of the study area is adjacent to the northern Jiangsu shoal, containing radial sand ridges. The special geographical location and climatic conditions make this area suitable for laver culture, and it has become the largest *P. yezoensis* culture base in China.

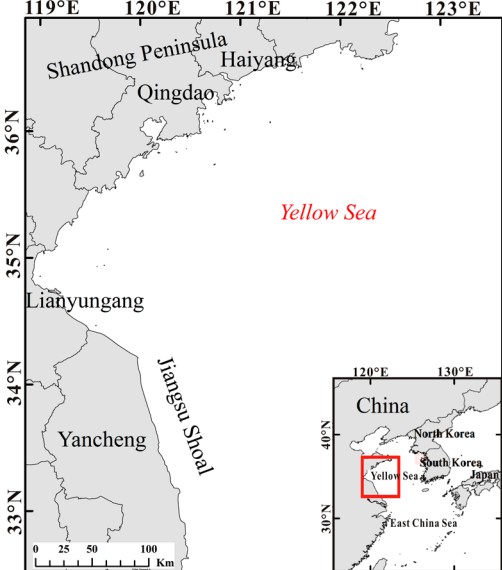

**Figure 1.** Map of the study area in the Southern Yellow Sea, China.

## 2.2. Remote Sensing Data and Processing

Operations, such as remote sensing, data selection, and preprocessing, were completed on the GEE. In this paper, the remote sensing image data were used included atmospherically corrected surface reflectance data from the Landsat8 OLI/TIRS sensors and Level-2A orthorectified atmospherically corrected surface reflectance data from Sentinel-2/MSI sensors. Landsat8/OLI images have a revisit period of 16 days and a spatial resolution of 30 m, with 11 bands, of which the eighth is a panchromatic band. The spatial resolution of Sentinel-2/MSI imagery is 10 m, 20 m, and 60 m, with 13 bands, and the revisit period is five days. The parameters of the satellite images selected in this paper are shown in Table 1. The reason for choosing these data was that, compared with high temporal resolution satellite data (such as MODIS and GOCI), the high spatial resolution satellite images were more in line with the actual situation. This reduces the effect of mixed pixels on the extraction of algae (especially in the period of early growth) and improves the accuracy of disaster monitoring.

**Table 1.** Satellite image information.

| Band | Wavelength (µm) Landsat8/OLI | Sentinel-2/MSI |
|------|------|------|
| 1 | 0.43–0.45 | 0.443 |
| 2 | 0.45–0.51 | 0.490 |
| 3 | 0.53–0.59 | 0.560 |
| 4 | 0.64–0.67 | 0.665 |
| 5 | 0.85–0.88 | 0.705 |
| 6 | 1.57–1.65 | 0.740 |
| 7 | 2.11–2.29 | 0.783 |
| 8 | 0.52–0.90 | 0.842 |
| 8A | - | 0.865 |
| 9 | 1.36–1.38 | 0.945 |
| 10 | 10.60–11.19 | 1.375 |
| 11 | 11.50–12.51 | 1.610 |
| 12 | - | 2.190 |
| Resolution | 30 m | 10 m |
| Revisit cycle | 16 days | 5 days |
| Date period | 2013-present | 2016–present |

"-" means no band.

Among them, the satellite images of the same date were crop and mosaic so as to transform the multi-scene small image into a large range of one scene image. The land mask was a mask operation performed after importing China's coastline data into GEE from 2016 to 2019. For the Landsat8/OLI images selected in this paper, the "Landsat. simpleCloudScore" cloud recognition and mask algorithm that comes with GEE was used. This algorithm added a "cloud" band to the image, with a band value of 0–100. The larger the value, the greater the possibility of clouds, so this paper set the value of the "cloud" band to 20. For the Sentinel-2/MSI images in the selected study area, this study used the QA60 band, which contains cloud information for cloud mask and image quality problems caused by cirrus clouds [34].

## 2.3. Field Data

Unlike the traditional method of verifying satellite accuracy based on the field measurement data, it is very difficult to directly validate the *U. prolifera* detection from satellite data using field measurements. This is because the macroalgae are in a wide range and patchy, and it is difficult to obtain near-timely field measurements of the wide distribution. Therefore, the field data were collected mainly to understand the spectral characteristics of the algae in order to help the identification of it in seawater (Figure 2). In situ reflectance of living floating macroalgae was measured on 30 June 2017 and 04 July 2019 when green tides arrived in coastal waters off Haiyang. A fiber-optic probe with a 10° field of view

connecting to a portable spectroradiometer (ASD FieldSpec) was pointed vertically 1 m over the sea surface to record the radiance of macroalgae ($L_a$, DN), and then pointed to a reference plaque with a calibrated reflectance of 0.25 to record the radiance of the plaque ($L_p$, DN); this procedure was repeated five times for every set of measurements to determine the mean $L_a$ and $L_p$. Reflectance of algae was calculated as $R$ (unitless) = $L_a \times 0.25/L_p$ [18].

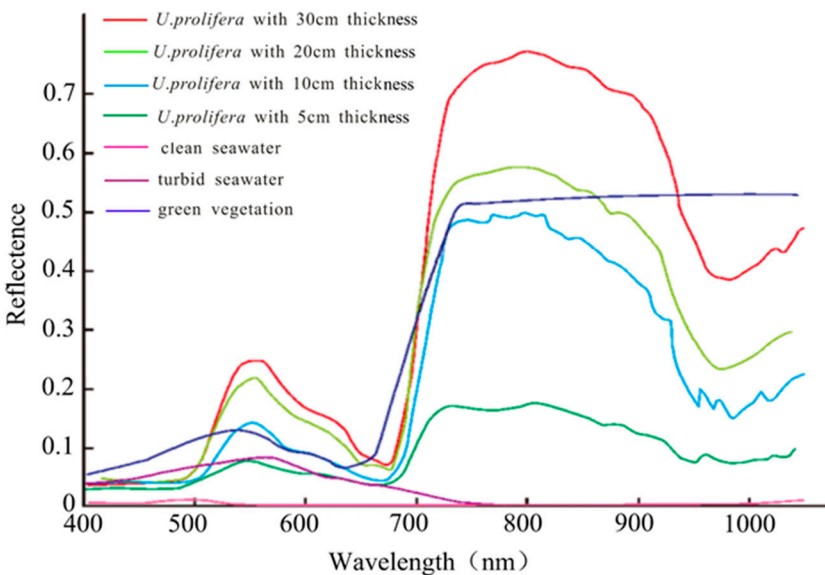

**Figure 2.** Reflectance spectra of the *U. prolifera* and seawater measured in situ.

*2.4. Methods*

2.4.1. Normalized Difference Vegetation Index

In the past few years, the Normalized Difference Vegetation Index (NDVI) has been widely used in the extraction and classification of algae [35–37]. The calculation formula of NDVI is as follows:

$$\text{NDVI} = (R_{NIR} - R_{RED})/(R_{NIR} + R_{RED}), \tag{1}$$

where $R_{NIR}$ and $R_{RED}$ represent the reflectance of the near-infrared and infrared bands, respectively, in the atmospheric window. In the visible and near-infrared bands, *U. prolifera* showed similar spectral characteristics to green vegetation, and the reflectance curve of algae increased sharply near 700 nm ("red edge"). The reflectance of seawater in this band is low, so it is easy to distinguish *U. prolifera* floating in seawater. Generally, the NDVI value is between −1 and 1, the NDVI value of the water body is less than 0, and the NDVI value of *U. prolifera* is greater than 0. However, the NDVI value in each scene image was not constant due to weather conditions. Therefore, it was not reasonable to set the NDVI threshold as 0 to distinguish the algae from the water body. In specific cases, it was necessary to set the threshold based on a large number of threshold experiments and manual visual interpretation results. The fifth and fourth bands of the Landsat8/OLI image and the eighth and fourth bands of Sentinel-2/MSI imagery were selected to calculate NDVI in this study.

2.4.2. Floating Algae Index

In GEE, Landsat8/OLI and Sentinel-2/MSI atmospheric correction data were converted to Rayleigh-corrected reflectance ($R_{rc}$) by the following formula:

$$R_{rc} = \frac{\pi L_t^*}{F_0 cos\theta_0} - R_r \tag{2}$$

where $L_t^*$ is the calibrated sensor radiance after adjustment for ozone and other gaseous absorption, $F_0$ is the extraterrestrial solar irradiance at data acquisition time, $\theta_0$ is the solar zenith angle, and $R_r$ is Rayleigh reflectance [38].

NDVI values fluctuate greatly and are sensitive to environmental changes, such as aerosol type and thickness, solar angle and observation geometry, and sun glint [39,40]. Unlike the green vegetation growing on the land, the seawater strongly absorbs the light in the short-wave infrared band, so the seawater appears "black" in this band, forming a strong contrast with *U. prolifera* floating on the sea [18,41]. Based on this, Hu et al. proposed a baseline subtraction algorithm, which can correct the atmosphere simply and effectively [23]. FAI is calculated as follows:

$$FAI = R_{NIR} - R'_{NIR} \tag{3}$$

$$R'_{NIR} = R_{RED} + (R_{SWIR} - R_{RED}) \times (\lambda_{NIR} - \lambda_{RED}) / (\lambda_{SWIR} - \lambda_{RED}), \tag{4}$$

where $R_{RED}$, $R_{NIR}$, and $R_{SWIR}$ represent the Rayleigh-corrected reflectance of the red, near-infrared, and short-wave infrared bands, respectively. $\lambda_{RED}$, $\lambda_{NIR}$ and $\lambda_{SWIR}$ represent the central wavelength of the red, near-infrared, and short-wave infrared bands, respectively. $R'_{NIR}$ represents the baseline reflectance of the near-infrared band obtained by linear interpolation between the red band and the short-wave infrared band. Compared with the NDVI algorithm, the FAI value has a smaller fluctuation range. However, the FAI algorithm separated floating algae from other nonbloom sea waters very well. This is understandable because at the bloom–nonbloom boundary, there should be a sharp change (large gradient) in the FAI values [42]. The FAI algorithm was initially applied to MODIS images. The central wavelengths of the red, near-infrared, and short-wave infrared bands of MODIS images were 645 nm, 859 nm, and 1240 nm. There are many bands of Landsat8/OLI and Sentinel-2/MSI data. In this paper, the sixth, fifth, and fourth bands of Landsat8/OLI images (central wavelengths were 1610 nm, 865 nm, and 655 nm, respectively) and the tenth, fifth, and fourth bands of Sentinel-2/MSI imagery (central wavelengths were 1375 nm, 705 nm, and 665 nm, respectively) were selected for FAI calculation in this paper [43–45].

### 2.4.3. Canny Edge Filter and Otsu Thresholding

As mentioned above, the index was calculated from the spectral characteristics of algae. However, due to the influence of the environment, sensors, and other factors, the spectral properties of algae were different, which made the calculated values of algae index different. Therefore, it was necessary to intervene in the threshold selection algorithm manually. The Otsu thresholding algorithm is the most widely used. It is used to automatically find the best threshold through the image histogram obtained by least squares [46–48]. In the Otsu algorithm, the optimal threshold was based on maximizing the interclass variance (equivalently, it minimizes the sum of intraclass variances), as in Equation (4):

$$BSS = \sum_{k=1}^{p} \left( \overline{V_k} - \overline{V} \right)^2, \tag{5}$$

where BSS represents the between-sum-of-squares, describing the variance structure of a dataset; $p$ is the number of classes, which in this study was 2. $V$ is the value of the band selected to divide different classes. Class $k$ is defined by every $V$ less than a certain threshold. The optimal threshold is obtained by maximizing the BSS [49].

The Canny Edge Filter was originally used to extract the boundary between rivers and the land. Here, the Canny Edge Filter was applied to the boundary between algae and seawater to improve the extraction accuracy of micro-*U. prolifera* scattered in the sea [50]. It should be noted that the distribution of the histogram appeared at the junction of the seawater and *U. prolifera* pixels. Therefore, the filter was buffered, and a buffer area of 10 m × 10 m was established at the edge [51]. In this paper, the parameters (sigma and threshold) of the Canny Edge Filter were set to $\sigma$ = 0.1, *th* = 0.01. The $\sigma$ and *th* parameters

were used to define the standard deviation of the Gaussian smoothing kernel and the sensitivity of the filter, respectively.

### 2.4.4. Model Building

In this paper, the adaptive threshold detection model, which enables fast and automated threshold selection by adding the Otsu threshold algorithm of the Canny Edge Filter based on algal calculation indices (FAI, NDVI), was used. It should be noted that this model only achieved a threshold when there was algae occurred in the seawater (we first perform FAI or NDVI calculation on the image before performing Canny Edge Filter and Otsu segmentation). When the model used the FAI algorithm, it was named FAI-COM (model based on FAI, the Canny Edge Filter, and Otsu thresholding). When the model used the NDVI algorithm, it was named NDVI-COM (model based on NDVI, the Canny Edge Filter, and Otsu thresholding). A demonstration of each module in the model is shown in Figure 3 (the gray line in Figure 3c represents the threshold only based on Index, and the red line represents the optimal threshold line based on Canny Edge Filter and Otsu method).

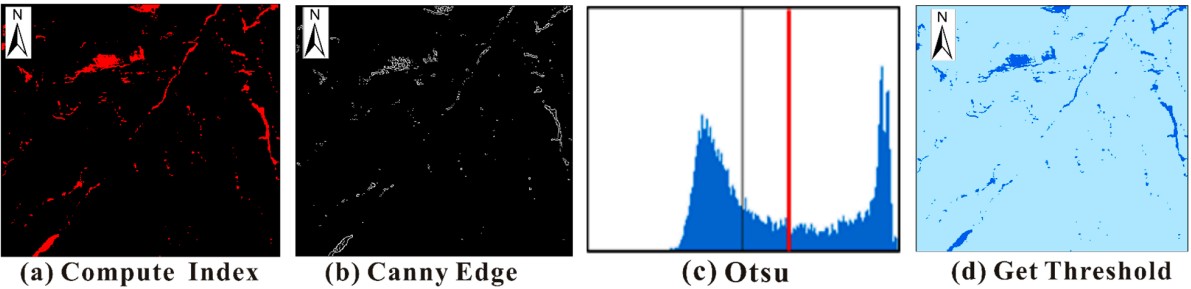

**Figure 3.** Demonstration of each algorithm in GEE.

The model was embedded into the GEE platform. After preprocessing the image data, the threshold values in different areas could be automatically selected, which greatly improved the efficiency of monitoring *U. prolifera* based on remote sensing methods. The technical route of this paper is shown in Figure 4. It should be noted that, after assessing the accuracy of the model, this paper determined the condition for using FAI-COM or NDVI-COM to generate *U. prolifera* results, and the condition was whether "the average cloud cover was <70% (for all images on the same date in the study area)". See Sections 3.1 and 4.1 for details.

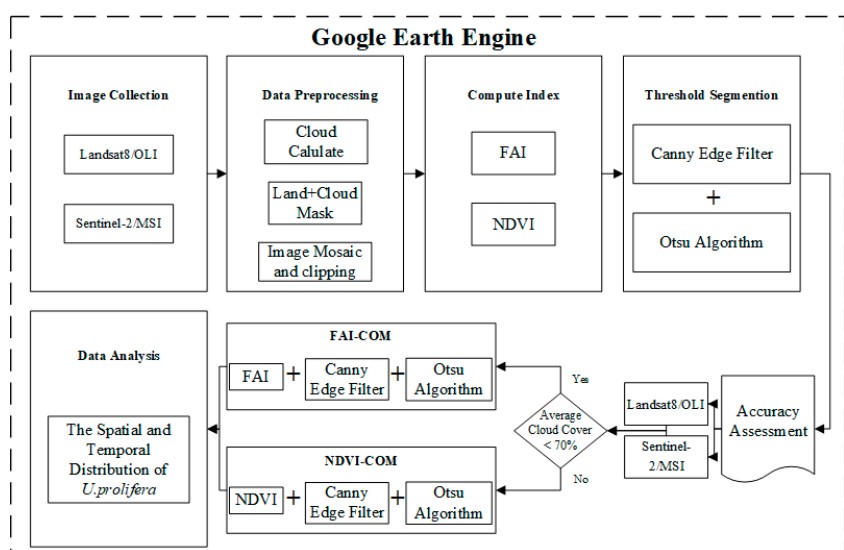

**Figure 4.** Workflow of adaptive threshold detection model.

### 3. Results

*3.1. Model Accuracy and Applicability*

3.1.1. Ground Truth

In this paper, we took the visual interpretation of macroalgae on the image as the ground truth for the model verification because the green macroalgae floating on the water surface has a generally similar spectral property to land vegetation in the visible and NIR wavelengths with a typical red-edge signal (700 nm). Moreover, based on previous research results, macroalgae disasters are often a single species in the study area. Although other algae, such as *Sargassum,* may be found in some years, the spectral reflectance characteristics of these algae are significantly different [52]. Therefore, the pixels of algae could be determined by comparing the spectral reflectance characteristics of satellite images (Figure 5) and in situ measured *U. prolifera* (Figure 2). In essence, visually determined algae slicks can be used as the ground truth [24].

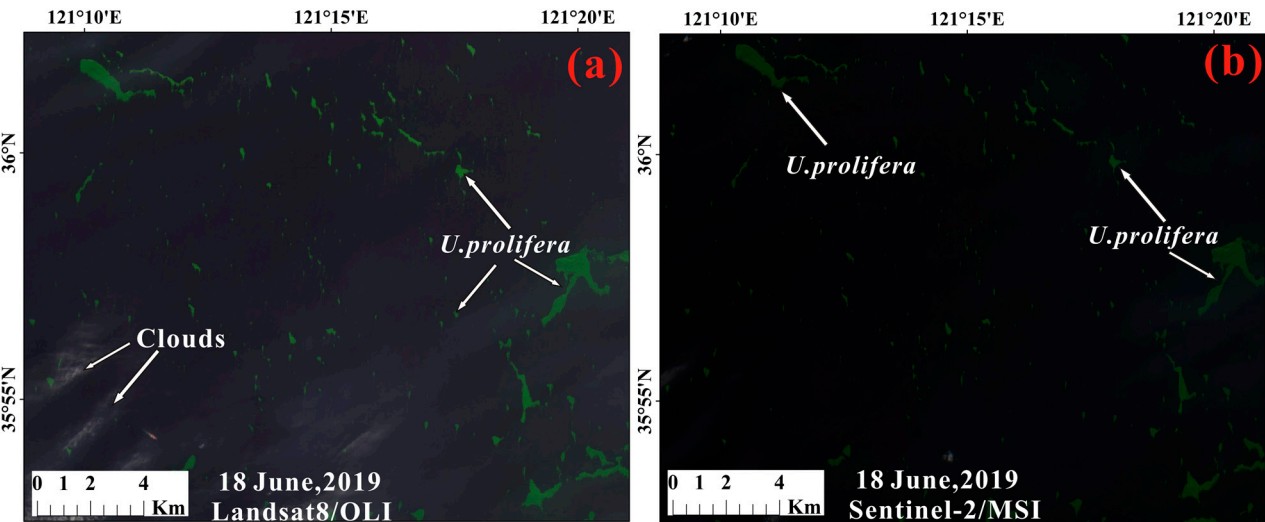

**Figure 5.** A visual interpretation of the satellite Red-Green-Blue "true-color" composite image recorded by Landsat8/OLI and Sentinel-2/MSI. (**a**) Landsat8/OLI true-color image on 18 June 2019, R:G:B = band 4:3:2, (**b**) Sentinel-2/MSI true-color image on 18 June 2019, R:G:B = band 4:3:2.

3.1.2. Accuracy Comparison Based on Cross-Sensor

Influenced by the environment and physiological characteristics, the spectral reflectance of *U. prolifera* is different in different periods [12,15,53]. In this study, we selected some images of *U. prolifera* at different scales and dates to verify the accuracy of the model. Compared with Landsat8/OLI image, Sentinel-2/MSI data have a short revisit period and high resolution (10 m). The method of artificial visual interpretation can be used to verify the accuracy of the model with Sentinel-2/MSI data.

On 3 June 2018, *U. prolifera* appeared red in the Sentinel-2/MSI false-color image. Then, combined with the spectral characteristics of random sampling points in Figure 6d, we found *U. prolifera* was mainly distributed in the sea area near the *P. yezoensis* culture area in the northern Jiangsu shoal and appeared sporadically at a small scale. The FAI-COM and NDVI-COM were calculated, and a comparison map of *U. prolifera* extraction was obtained, as shown in Figure 6b,c. As shown in the red circles, the *U. prolifera* prevalence extracted based on FAI-COM was more consistent with the distribution of *U. prolifera* in the image, while there were more algae pixels extracted by NDVI-COM (*N* = 207,164, "N" means the number of *U. prolifera* pixels) than by FAI-COM (*N* = 199,731), which was inconsistent with the actual situation.

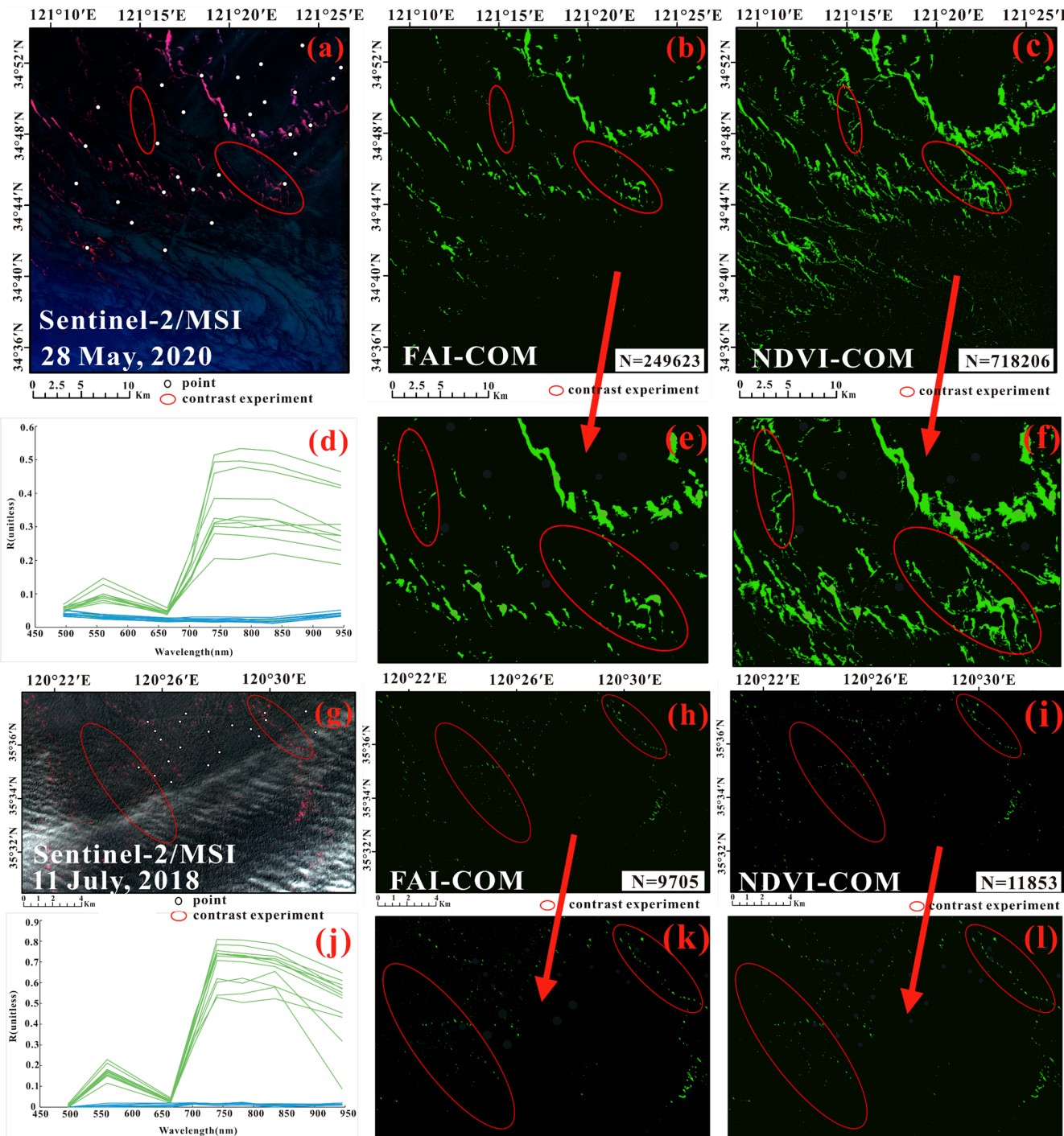

**Figure 6.** Comparison of *U. prolifera* extraction from Sentinel-2/MSI images on 28 May 2020 (**a**–**f**) and 11 July 2018 (**g**–**l**). "N" means the number of *U. prolifera* pixels. (**a**,**g**) Sentinel-2/MSI pseudo-true-color image, R:G:B = band 8:4:3, (**b**,**h**) *U. prolifera* extracted from FAI-COM, (**c**,**i**) *U. prolifera* extracted from NDVI-COM, (**d**) reflectance spectra with random points of *U. prolifera* and seawater corresponding to the eight Sentinel-2 bands on 28 May 2020, (**e**,**f**) enlarged view of (**b**,**c**), respectively, (**j**) reflectance spectra with random points of *U. prolifera* and seawater corresponding to the eight Sentinel-2 bands on 11 July 2018, (**k**,**l**) enlarged view of (**h**,**i**), respectively.

On 11 July 2018, according to the spectral characteristics of random sampling points and visual interpretation of macroalgae in Figure 6, it was found that the *U. prolifera* were scarce and distributed sporadically in the south of Qingdao City, Shandong Province. FAI-COM and NDVI-COM were calculated for this image. As shown in Figure 6g, there were a

few thin clouds. However, *U. prolifera* were distributed under thin clouds. In the red circles, the extracted distribution of algae based on FAI-COM (*N* = 9705) was more consistent with the distribution of *U. prolifera* in this image, and the number of *U. prolifera* pixels was less than that extracted by NDVI-COM (*N* = 11,853). The image of thin cloud pixel also had a higher NDVI value, which increased the extraction number of *U. prolifera* pixels.

Comparing the extraction of *U. prolifera* in the red circles of Figure 6, it can be found that the model based on the NDVI algorithm (NDVI-COM) was classified into pure pixels of *U. prolifera* by mistake under the same parameters, which led to the overall number of *U. prolifera* pixels being more than that of FAI-COM. This phenomenon exists in remote sensing images of *U. prolifera* at different scales and dates. Due to the influence of satellite image quality, atmospheric environment conditions (thin clouds, sun glint), and satellite resolution, NDVI-COM is affected by environmental changes and has poor robustness. Therefore, the model based on the FAI algorithm (FAI-COM) is more suitable for extracting *U. prolifera* from the Sentinel-2/MSI image.

As mentioned above, this paper takes the Sentinel-2/MSI image data with higher spatial resolution as the true values and uses them to perform a cross-sensor comparison on Landsat8/OLI image data of relatively low resolution. From 2016 to 2020, there were four scenes of Landsat8/OLI and Sentinel-2/MSI images with the same date in the study area, as shown in Table 2.

**Table 2.** Landsat8/OLI and Sentinel-2/MSI image data.

| Date (YYYYMMDD) | Landsat8/OLI Image Time | Sentinel-2/MSI Image Time |
|---|---|---|
| 20160616 | 02:36:16 UTC | 02:45:52 UTC |
| 20170628 | 02:29:45 UTC | 02:35:51 UTC |
| 20190618 | 02:29:26 UTC | 02:35:51 UTC |
| 20190713 | 02:23:21 UTC | 02:35:59 UTC |

The width of the Landsat8/OLI image is unlike that of the Sentinel-2/MSI image. For better accuracy, different regions of interest were randomly selected from the overlapping regions of two kinds of images. The distribution of regions of interest (ROI) is shown in Figure 7. It should be noted that on 13 July 2019, there were a large number of clouds in the study area, and there were few *U. prolifera* in the Landsat8/OLI images on that day, so only one region of interest was selected.

Next, the above data were processed. As shown in Figure 8, the extraction comparison map of interest area was obtained by the model (FAI-COM and NDVI-COM).

Figure 8 reveals that, similar to the Sentinel-2/MSI imagery on the same date, the Landsat8/OLI imagery on 16 June 2016 also had sun glint. The distribution of *U. prolifera* from FAI-COM (*N* = 27,418) was closer to the distribution of the *U. prolifera* in the actual image. The distribution of *U. prolifera* extracted by NDVI-COM (*N* = 71,174) was misclassified in the pixels because of the sun glint, which made the overall classification of *U. prolifera* pixels increase and not conform to the actual situation. This phenomenon was similar to the misclassification of *U. prolifera* in Sentinel-2/MSI images (Figure 6a).

This paper analyzed the correlation between the area of *U. prolifera* extracted from the two algorithm models (Figure 9). The correlation results showed that, through study area thresholds, the area of *U. prolifera* extracted by two models were well correlated (greater than 0.9), but the root mean square error of FAI-COM was small (RMSE = 5.64).

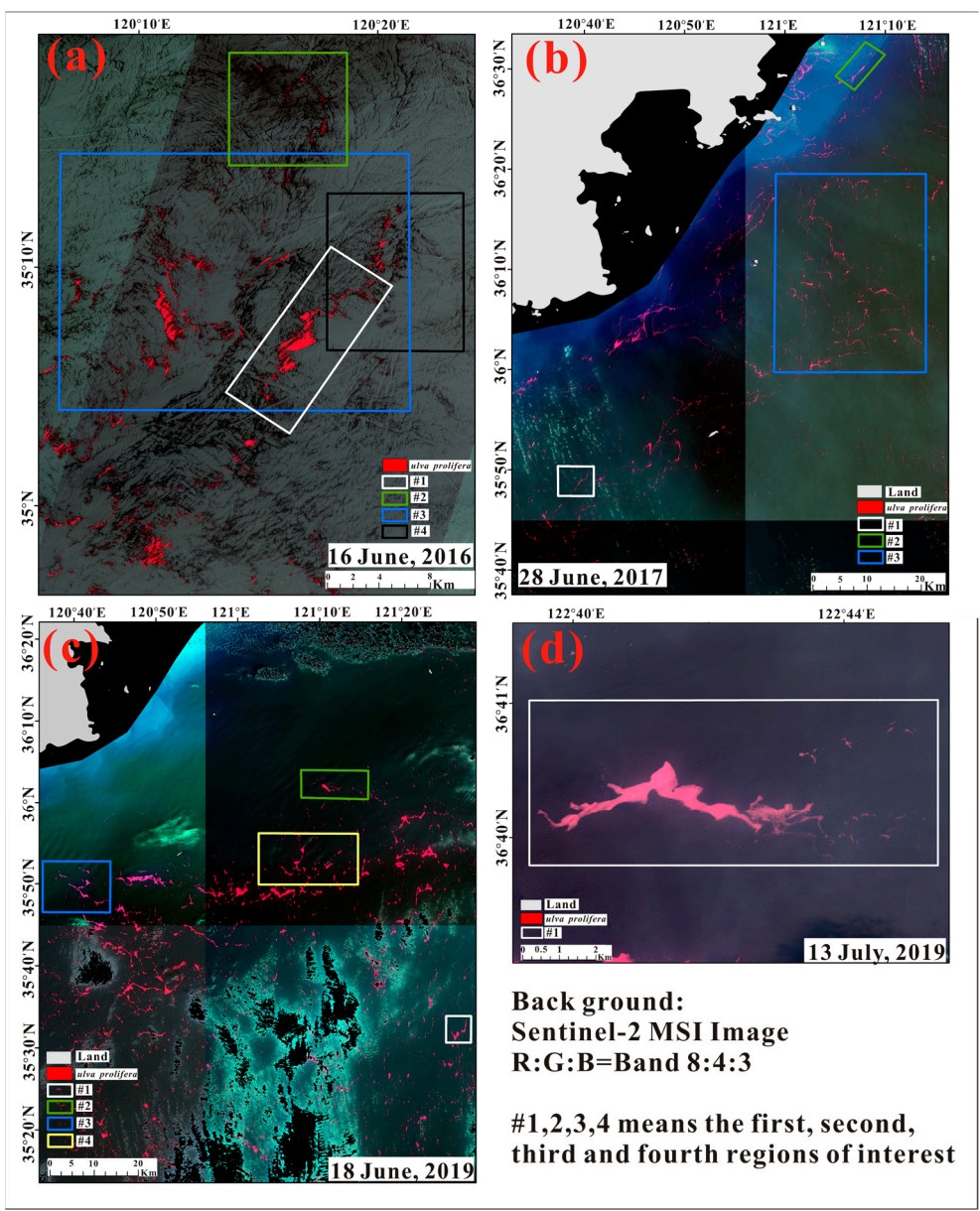

**Figure 7.** Map of regions of interest. (**a**) Distribution map of four ROI on 16 June 2016, (**b**) Distribution map of three ROI on 28 June 2017, (**c**) Distribution map of four ROI on 18 June 2019, (**d**) A distribution map of one ROI on 13 July 2019.

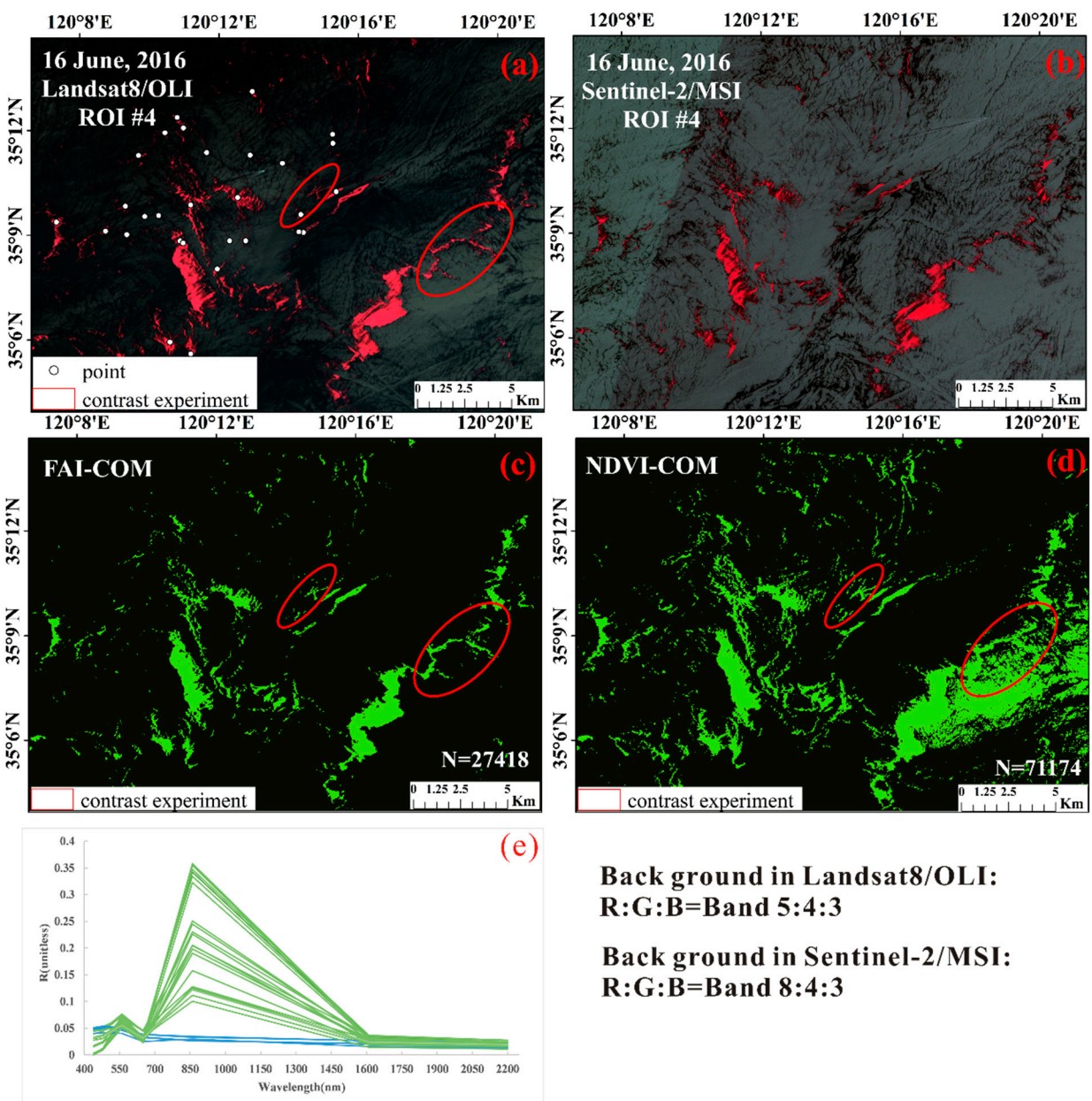

**Figure 8.** Comparison of *U. prolifera* extraction in #4 ROI on 16 June 2016. "N" means the number of *U. prolifera* pixels. (**a**) Landsat8/OLI pseudo-true-color image, R:G:B = band 5:4:3, (**b**) Sentinel-2/MSI pseudo-true-color image, R:G:B = band 8:4:3, (**c**) *U. prolifera* extracted from FAI-COM, (**d**) *U. prolifera* extracted from NDVI-COM, (**e**) reflectance spectra with random points of *U. prolifera* and seawater corresponding to the seven Landsat8 bands on 16 June 2016.

Overall, the accuracy of FAI-COM was higher in Landsat8/OLI and Sentinel-2/MSI images, but when there were more clouds (such as 13 July 2019) in the imagery, FAI-COM was no longer applicable. Therefore, the average cloud cover of the image was calculated in the study area from 2016 to 2020, and then the cloud cover information of the date was counted when FAI-COM was not applicable to extract algae (Table 3).

Based on the statistical results of the average cloud cover of images in Table 3, the condition of using FAI-COM or NDVI-COM was set to 70% of the average cloud cover of all images on the same date. Then, the FAI-COM was used to generate *U. prolifera* results in the study area when the average cloud cover was less than 70% (model parameter: $\sigma = 0.1$, $th = 0.01$); on the contrary, NDVI-COM was used to generate *U. prolifera* results when the average cloud cover more than 70% (model parameter: $\sigma = 0.1$, $th = 0.1$). See Section 4.1.1 for detailed analysis.

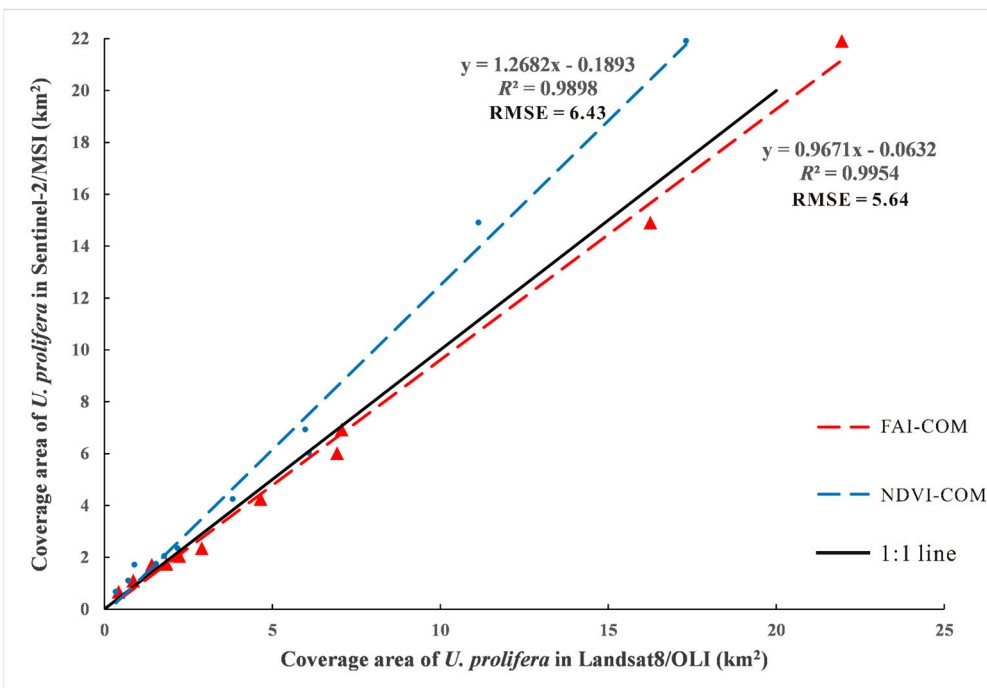

**Figure 9.** Accuracy comparison of two models for extracting *U. prolifera* in Landsat8/OLI images. The coverage area of *U. prolifera* in Sentinel-2/MSI is taken as the true value.

**Table 3.** Statistical table of cloud cover information in the date when FAI-COM was not able to extract *U. prolifera*.

| Date (YYYYMMDD) | Cloud Cover | | | | Average Cloud Cover |
|---|---|---|---|---|---|
| 20170612 **** | 98.37% | 91.65% | 87.44% | 87.35% | 91.20% |
| 20170619 *** | 93.23% | 74.6% | 66.99% | | 78.27% |
| 20180622 *** | 96.65% | 96.3% | 94.22% | | 95.72% |
| 20180701 **** | 93.27% | 81.25% | 78.38% | 73.92% | 81.71% |
| 20190627 *** | 93.79% | 84.56% | 62.74% | | 80.36% |
| 20190713 ** | 78.79% | 64.15% | | | 71.47% |

"****" means that there were four scenes of images in the study area during this date; "***" means that there were three scenes of images, and "**" means that there were two scenes of images in the study area.

### 3.1.3. Accuracy Comparison Based on Confusion Matrix

A total of 15 Sentinel-2/MSI images from 2016 to 2020 were selected as training samples, as well as for evaluation of the confusion matrix of the two models (FAI-COM and NDVI-COM) used in Landsat8/OLI images. A confusion matrix is a common method for evaluating the accuracy of two or more classes of classification and is often used to evaluate the performance of classification methods. Based on the confusion matrix, the overall accuracy (OA), user accuracy (User acc.), F1 score, and Kappa coefficients were calculated [54–56]. The classification of 15 test images was determined by the algorithm index and visual interpretation experience. These independent images with a representative environment (clear sky, thin clouds) and aggregate conditions were used as training samples. The statistical table of the confusion matrix of the model is shown in Table 4, where the overall classification accuracy of the model was above 0.9 with a high kappa coefficient value. The F1 score of the *U. prolifera* in the confusion matrix was above 0.8 overall. These results showed that the F1 score of the *U. prolifera* was lower than 0.8 on 12 June 2017, 19 June, and 22 June 2018, and 27 June, 1 July, and 13 July 2019. On these days, there were a lot of clouds in the Landsat8/OLI images (Table 3), and only a small amount of *U. prolifera* was distributed at the edge of the cloud. In these circumstances, NDVI-COM was able to classify *U. prolifera*. In addition, the resolution of Landsat8/OLI images was

lower than the Sentinel-2/MSI sample data, and NDVI-COM classified more mixed pixels of algae, which led to the lower F1 scores of the model.

**Table 4.** Confusion matrix of FAI-COM and NDVI-COM.

| Date | OA | Kappa Coefficient | User acc. | F1 Score |
|------|-----|------------------|-----------|----------|
| 20160524 | 0.98 | 0.92 | 98.6 | 0.93 |
| 20160609 | 0.98 | 0.8675 | 97.41 | 0.89 |
| 20160616 | 0.98 | 0.9356 | 98.57 | 0.94 |
| 20160625 | 0.99 | 0.9681 | 94.12 | 0.96 |
| 20160702 | 0.99 | 0.8863 | 99.61 | 0.88 |
| 20160711 | 0.97 | 0.7124 | 96.68 | 0.92 |
| 20160718 | 0.99 | 0.8352 | 97.92 | 0.94 |
| 20170527 | 0.99 | 0.8872 | 96.6 | 0.89 |
| 20170612 * | 0.98 | 0.6509 | 49.14 | 0.76 |
| 20170619 * | 0.99 | 0.9222 | 50.69 | 0.72 |
| 20170628 | 0.99 | 0.8454 | 83.52 | 0.85 |
| 20170705 | 0.99 | 0.8444 | 89.14 | 0.94 |
| 20180615 | 0.97 | 0.9371 | 78.69 | 0.85 |
| 20180622 * | 0.99 | 0.6743 | 51.19 | 0.68 |
| 20180701 * | 0.99 | 0.631 | 46.38 | 0.63 |
| 20190602 | 0.99 | 0.8676 | 84.81 | 0.87 |
| 20190618 | 0.99 | 0.8613 | 88.68 | 0.86 |
| 20190627 * | 0.99 | 0.8003 | 91.26 | 0.7 |
| 20190704 | 0.98 | 0.8395 | 83.59 | 0.94 |
| 20190711 | 0.98 | 0.8159 | 77.58 | 0.92 |
| 20190713 * | 0.98 | 0.728 | 85.38 | 0.73 |
| 20190720 | 0.99 | 0.9023 | 84.25 | 0.9 |
| 20190805 | 0.99 | 0.839 | 77.55 | 0.84 |
| 20200604 | 0.99 | 0.9165 | 84.87 | 0.92 |
| 20200611 | 0.99 | 0.8643 | 86.44 | 0.87 |
| 20200620 | 0.99 | 0.9391 | 92.31 | 0.94 |

"*" means model used NDVI (NDVI-COM).

We applied this model to Landsat8/OLI and Sentinel-2/MSI images and then performed *U. prolifera* extraction. The F1 score of *U. prolifera* extracted by the combined model was above 0.85. The results showed that the model had high extraction accuracy and strong robustness to environmental changes, as well as better applicability.

### 3.2. Spatio-Temporal Distribution of U. prolifera

Based on the adaptive threshold calculated by the model, the coverage area and spatiotemporal distribution of *U. prolifera* in the study area over five years were generated from Landsat8/OLI and Sentinel-2/MSI images (Figures 10 and 11).

On 25 June 2016, *U. prolifera* had the largest scale and the widest distribution range (from the eastern sea area of Yancheng City, Jiangsu Province to the southern sea area of Weihai City, Shandong Province). The daily coverage area of *U. prolifera* was about 1582 km$^2$, which was the largest coverage area of *U. prolifera* in the five years. In general, 2016 was the largest outbreak of *U. prolifera* in the five years. Previous studies have shown that 2015 was the most serious year of the *U. prolifera* disaster in the study area. After that, in 2016, the coverage area of *U. prolifera* decreased, but the distribution scope and duration persisted [57–59].

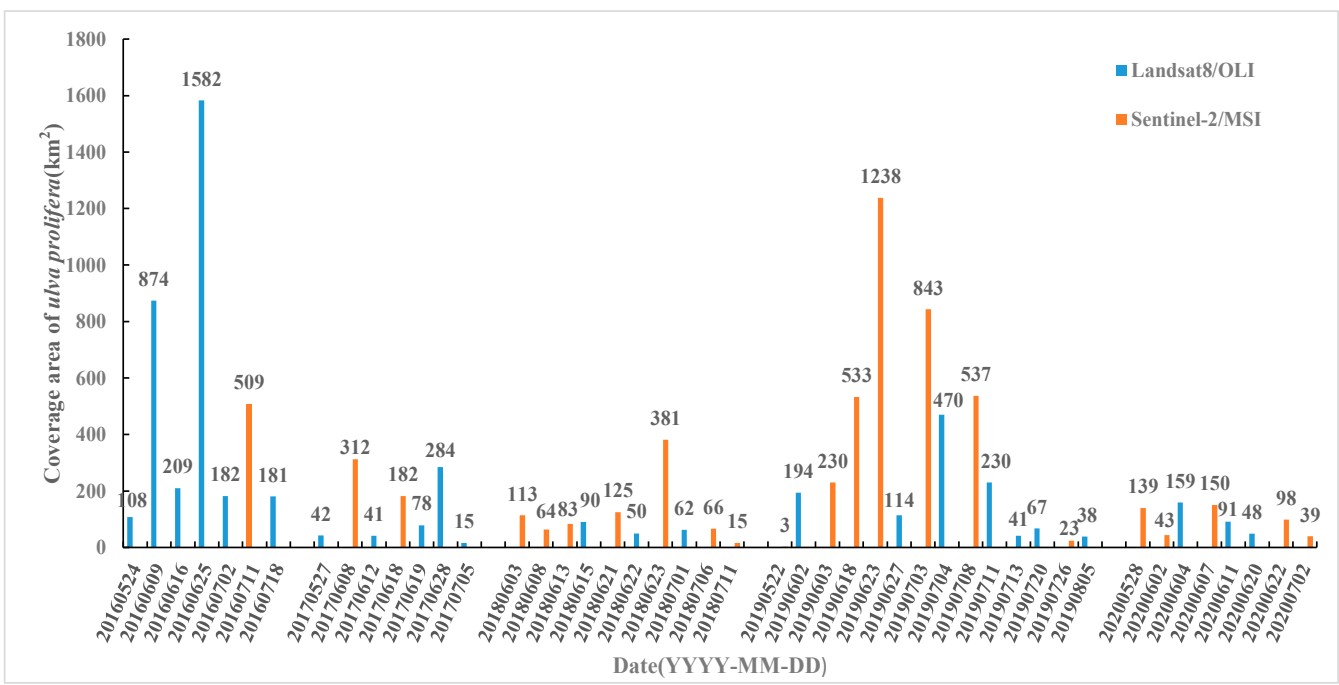

**Figure 10.** Statistical chart of *U. prolifera* area based on Landsat8/OLI and Sentinel-2/MSI satellites in the study area from 2016 to 2020.

Compared with 2016, the daily coverage area of *U. prolifera* in 2017 showed a significant downward trend. On 28 June 2017, *U. prolifera* had the largest scale, mainly distributed in the sea area near the Shandong Peninsula (Qingdao City, Yantai City, and Weihai City), with a daily coverage area of about 284 km$^2$. In 2016, the first discovery date of *U. prolifera* was 24 May; it covered an area of 108.07 km$^2$ in the northeast sea area of Yancheng City, Jiangsu Province. In 2017, the first discovery date of *U. prolifera* was 27 May; it covered an area of 39.59 km$^2$, and the sea area where *U. prolifera* gathered was also near Yancheng City, Jiangsu Province. The largest area was on 28 June in that year, with a daily coverage of about 284 km$^2$. *U. prolifera* was mainly distributed in the sea area near Yantai City with Sentinel-2/MSI data. There was no rebound in the outbreak scale of *U. prolifera* in 2018. Sentinel-2/MSI images showed that on 21 June 2018, *U. prolifera* had the largest scale, and was mostly distributed in the southern sea area of Qingdao City, and a small part was in the sea area near Yancheng City. The daily coverage area of *U. prolifera* was about 125 km$^2$. In 2018, *U. prolifera* was first found on 3 June (Sentinel-2/MSI image). At this time, *U. prolifera* was distributed in the eastern sea area of Yancheng City, Jiangsu Province, with daily coverage of about 113 km$^2$. Affected by the weather conditions in 2018 and 2020, Landsat8/OLI images only monitored the distribution of *U. prolifera* in June. The outbreak scale in these two years was much smaller than that in 2016.

In 2019, the outbreak scale of *U. prolifera* increased compared to the previous three years. On 23 June, *U. prolifera* broke out at the largest scale. Through Sentinel-2/MSI images, it was shown that algae were distributed in most of the sea areas of Shandong Peninsula and Yancheng City, Jiangsu Province, with a daily coverage of about 1238 km$^2$. The outbreak scale in 2019 was like that in 2016. On 22 May, a Sentinel-2/MSI pseudo-true-color image showed *U. prolifera* floating in the sea area near the northeast of Yancheng City. After that, algae were detected again on 2 June, when the coverage area was about 194 km$^2$.

It should be noted that the Landsat8/OLI and Sentinel-2/MSI data cannot cover the whole study area. Therefore, in order to reflect the trend of the distribution area of *U. prolifera* more accurately over the past five years, MODIS data were selected in this paper. Then preprocessed MODIS Level-1A images, such as radiation correction, atmospheric correction, and geometric correction, were extracted from the distribution area of *U. prolifera*

by artificial visual interpretation. The MODIS results (Figure 12) were similar to the scale of *U. prolifera* reported by the Ministry of Natural Resources North China Sea Administration.

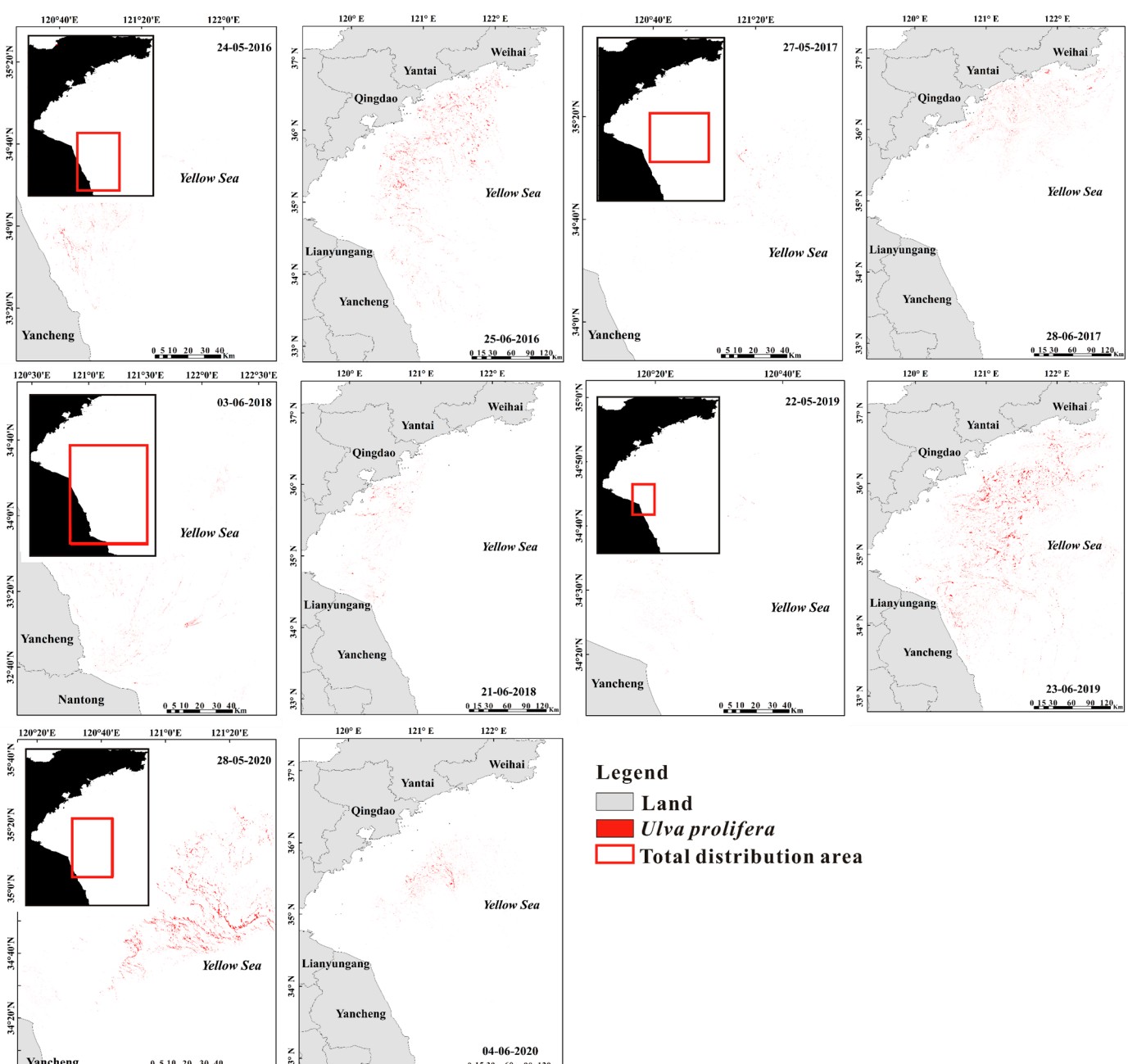

**Figure 11.** Temporal and spatial distribution of *U. prolifera* in the study area from 2016 to 2020. Only two representative images with the earliest discovery date and the largest daily area of *U. prolifera* were selected for each year.

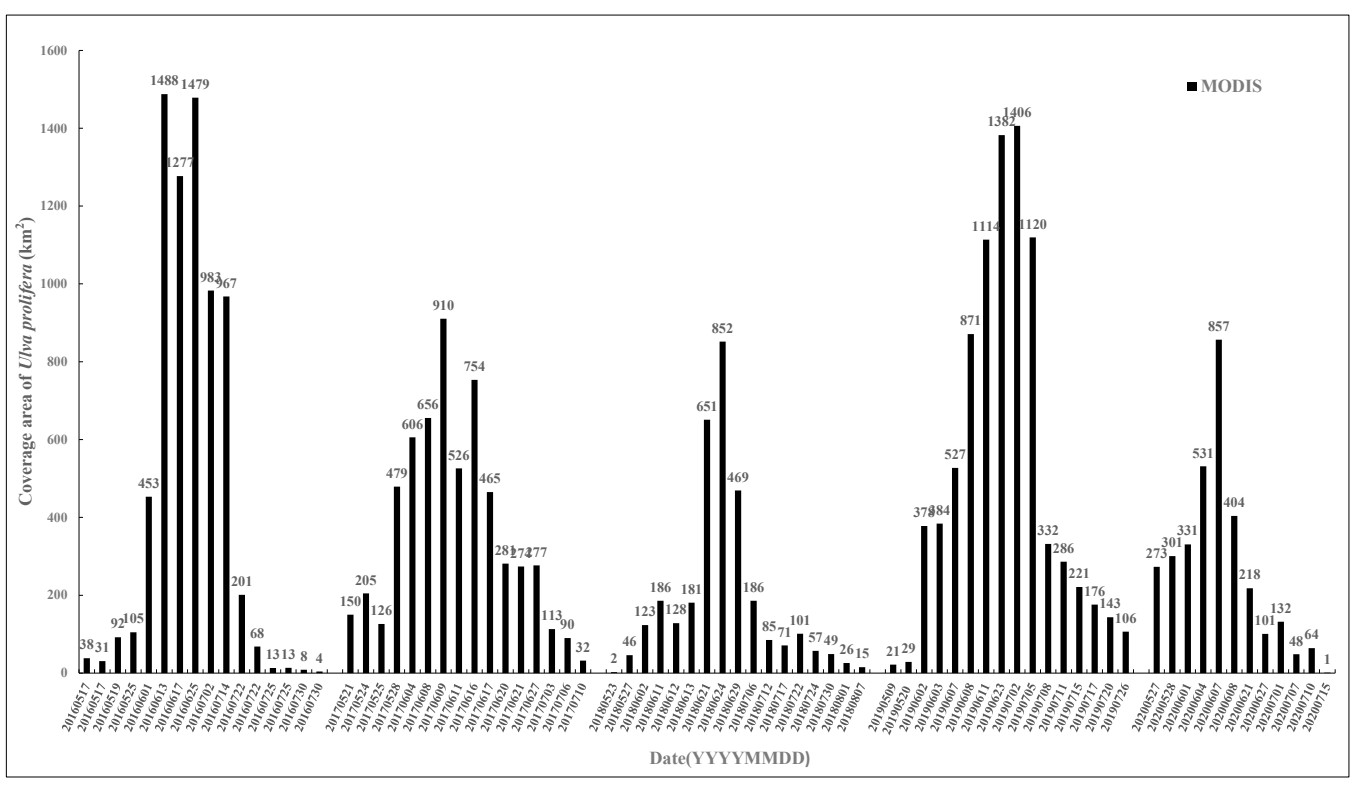

**Figure 12.** Statistical chart of *U. prolifera* area based on MODIS satellite in the study area from 2016 to 2020.

Through the annual average statistics of the coverage area of *U. prolifera* in the five years (Figure 13), it was found that the area showed a decreasing trend year by year. And the results of MODIS were consistent with those of Landsat8/OLI and Sentinel-2/MSI. In general, the interannual coverage of *U. prolifera* in 2017, 2018, and 2020 was relatively small, while that of 2016 and 2019 was relatively large. Among the Landsat8/OLI and Sentinel-2/MSI results, the annual average area of algae in 2016 was the largest (521 km$^2$); compared with 2017 and 2018, the annual mean coverage of algae in 2019 increased to about 326 km$^2$.

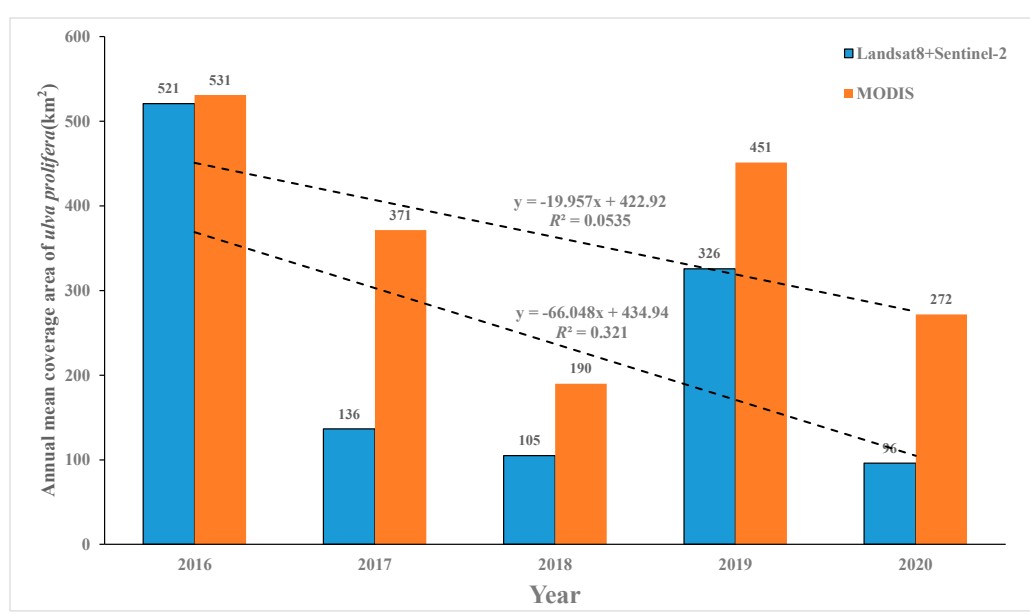

**Figure 13.** Statistical chart of annual mean coverage area of *U. prolifera*.

Compared to Figure 10, the daily coverage area of *U. prolifera* in Figure 12 was larger, which was related to the existence of many mixed pixels in MODIS images (many pixels may be mixed with both water and macroalgae). It was also related to the smaller width of Landsat8/OLI and Sentinel-2/MSI satellites, which did not cover the whole area. But the area of *U. prolifera* in the early stage (late May to early June) over the five years in Figure 12 was similar to that in Figure 10. By contrast, high-resolution satellite images were more suitable for the extraction of the early stage of the algae because the coverage area of *U. prolifera* was small and scattered in patches in this stage. In addition, the largest daily coverage area extracted by the model in this paper was similar to the data of the China Marine Disasters Bulletin from 2008 to 2019. Overall, the combined results of Landsat8/OLI and Sentinel-2/MSI can support the monitoring and early warning of *U. prolifera* in the study area.

## 4. Discussion

### 4.1. Evaluation of the Model

#### 4.1.1. Advantages of the Model

(1) The advantage of the adaptive threshold extraction model is that it can select the threshold automatically for the whole research area. The model integrates the advantages of the FAI algorithm and has high stability. That is, when the dynamic threshold calculated from a large area was applied to the extraction of *U. prolifera* in a small area, the extraction result was similar to the actual distribution area of *U. prolifera*.

Therefore, this paper selected four Landsat8/OLI images with repeated dates in Section 3.1, used the thresholds obtained in the whole study area to calculate the extraction area of *U. prolifera* in the ROI (small area), and then compared the stability of the model (Figure 14). It should be noted that the dynamic threshold model could extract the threshold in different ranges and had high accuracy (as shown in Figure 9). Therefore, the true value in Figure 14 was the area of *U. prolifera*, extracted by the dynamic threshold model based on the ROI in Sentinel-2/MSI images.

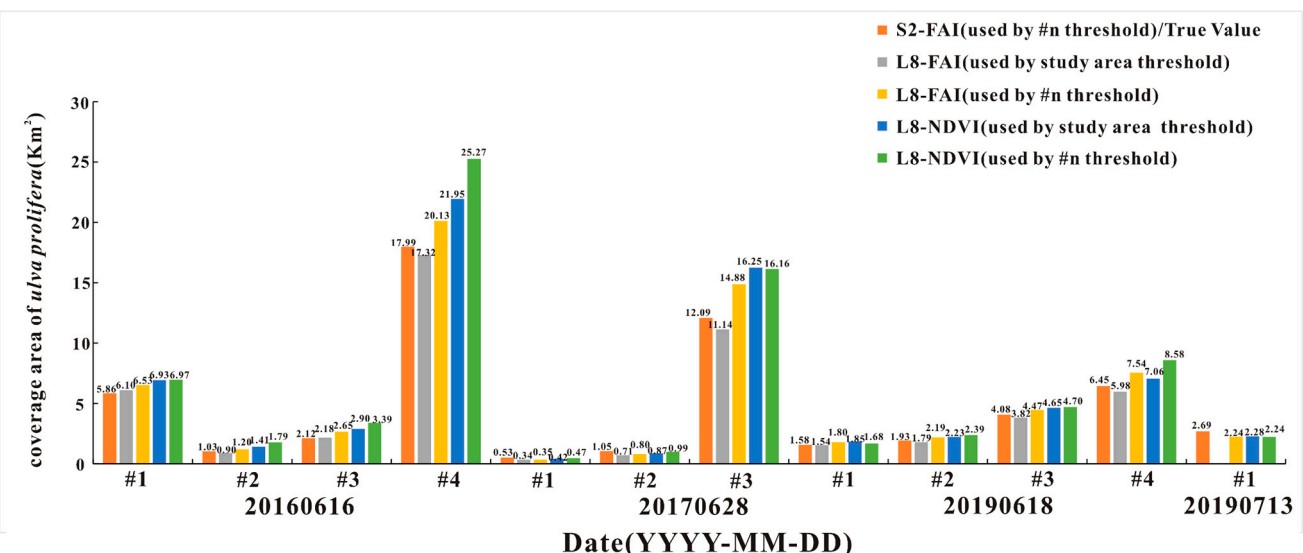

**Figure 14.** Statistical map of *U. prolifera* area. "S2" represents Sentinel-2/MSI images, "L8" represents Landsat8/OLI images, "used by #n threshold" represents the area of *U. prolifera* based on the threshold of the *n*-th ROI, *n* = 1, 2, 3, 4, "used by study area threshold" represents the area of *U. prolifera* based on the threshold of the whole study area.

It can be seen from Figure 14 that, based on NDVI-COM, under the threshold of the whole study area, the extracted area of *U. prolifera* was larger than that of the Sentinel-2/MSI image on the same date. Based on FAI-COM, the area of *U. prolifera* from the whole study area was similar to the true area of *U. prolifera,* and the area change was small. Therefore,

the adaptive threshold model based on FAI is more stable and accurate. This is mainly because the value fluctuation range of FAI is small (Figure 15), which makes the area of *U. prolifera* extracted between different thresholds change less.

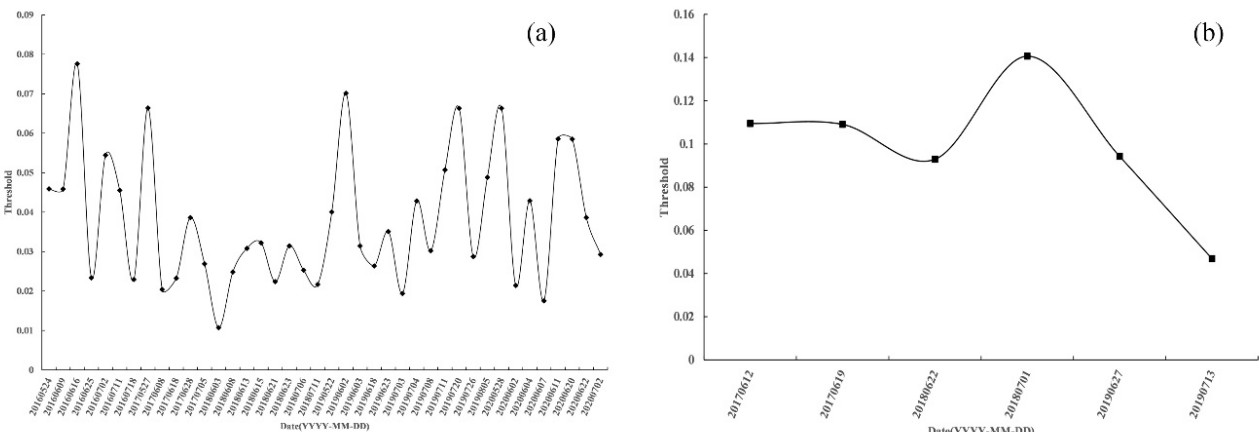

**Figure 15.** Statistical of optimal threshold of *U. prolifera* in the study area from 2016 to 2020: (**a**) threshold by FAI-COM; (**b**) threshold by NDVI-COM.

(2) By adjusting the parameters of the Canny Edge Filter (*th* = 0.1), the NDVI-COM realized the adaptive threshold extraction of *U. prolifera* when the average cloud cover was more than 70%. This is because, in the NDVI algorithm, the values of cloud and seawater were generally negative [60,61]. The parameter *th* of the Canny Edge Filter can determine the minimum gradient magnitude, and the sigma parameter ($\sigma$) is the standard deviation (SD) of a Gaussian prefilter to remove high-frequency noise. When *th* was set to a larger value (such as 0.1, the minimum gradient magnitude was similar to the difference in NDVI value between cloud and algae), the model could extract algae at the edge of thick clouds.

As mentioned above, the average cloud cover of Langsat8/OLI image on 13 July 2019 was higher than 70%. As shown in Figure 16, the cloud mask algorithm provided by GEE could not mask the cloud well, and a small part of algae was distributed at the edge of the cloud. After calculating the FAI and NDVI of the image, we found that the remaining cloud had a high FAI value (the maximum is about 0.06), while the NDVI value was negative. The higher FAI value of cloud could affect the threshold selection of the model, which made the FAI-COM no longer applicable at this time.

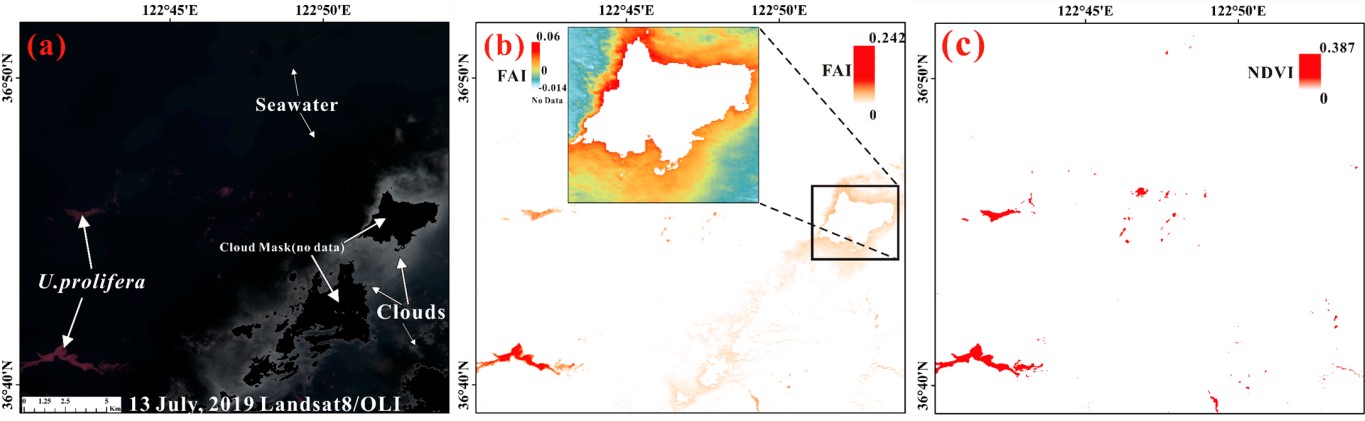

**Figure 16.** Schematic diagram of the model. (**a**) Landsat8/OLI pseudo-true-color image on 13 July 2019, R:G:B = band 5:4:3, (**b**) the result of FAI value greater than 0 on 13 July 2019, and the black box is the result of FAI value in cloud area, (**c**) the result of NDVI value greater than 0 on 13 July 2019.

(3) Another advantage of the dynamic threshold is that, compared with the fixed threshold, the extracted area of *U. prolifera* was more in line with the actual situation. Using the threshold extraction method by Hu et al. [42], this paper first calculated the mean and standard deviation by all individual images. Then, the fixed threshold was obtained by the mean minus two standard deviations. As shown in Figure 17b–d, in the sea area near Yantai City, Shandong Province on 25 June 2016, the fixed FAI threshold (T = 0.0044, Mean threshold value = 0.0378, and Stdev threshold value = 0.0167) and the fixed NDVI threshold (T = 0.0374, Mean threshold value = 0.0988, and Stdev threshold value = 0.0307) were not suitable for the extraction of *U. prolifera* while the dynamic threshold model in this paper could accurately extract *U. prolifera*. We found that the number of algae pixels extracted by the fixed threshold method ($N_{FAI}$ = 2813602 and $N_{NDVI}$ = 2640783) were more than the model proposed in this paper ($N_{Model}$ = 2149692), and it was not consistent with the distribution of *U. prolifera* in Figure 17a.

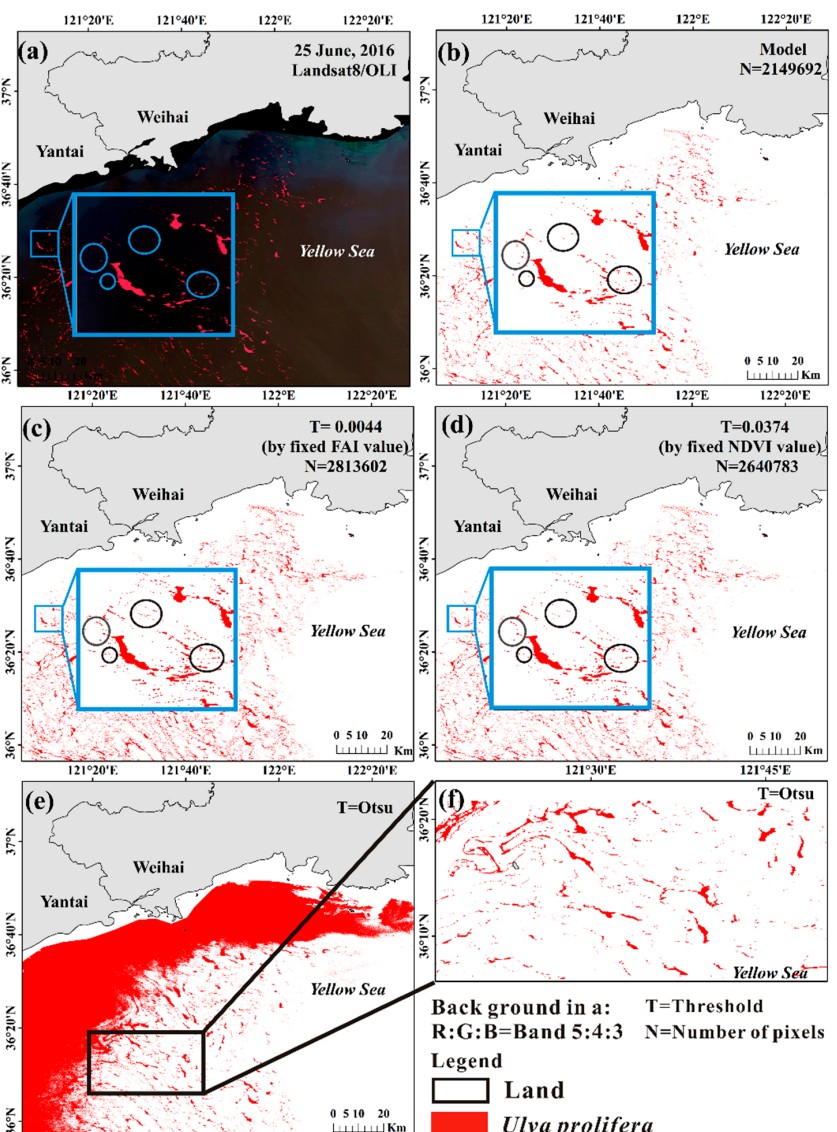

**Figure 17.** Comparison of adaptive threshold model with other methods. The blue box is the zoom of the same area in (**a**–**d**). (**a**) Landsat8/OLI pseudo-true-color image on 25 June 2016, R:G:B = band 5:4:3, (**b**) distribution map of *U. prolifera* based on the model in this paper, (**c**) distribution map of *U. prolifera* based on a fixed FAI threshold of 0.0044, (**d**) distribution map of *U. prolifera* based on a fixed NDVI threshold of 0.0374, (**e**) distribution map of *U. prolifera* based on the Otsu method, (**f**) distribution map of *U. prolifera* based on the Otsu method from the black box area in (**e**).

The Otsu threshold algorithm had defects in extracting *U. prolifera* on 25 June 2016, as shown in Figure 17e,f. The results showed that the Otsu algorithm was not suitable for large-scale threshold selection because the number of seawater pixels was far more than that of *U. prolifera*. When the research area was reduced, the Otsu algorithm showed better threshold selection results (Figure 17f). Compared with the Otsu algorithm, the dynamic threshold model in this paper was more suitable for a wide study area of threshold selection, which was related to the Canny Edge Filter added to the model. This filter improved the defects of the Otsu algorithm so that it only classified the edge pixels of *U. prolifera* and seawater by a histogram, limiting the area of the Otsu algorithm [62].

4.1.2. Uncertainty of the Model

(1) The construction of the model was mainly based on the GEE. The advantage of the GEE lies in its fast and convenient cloud processing operation, which saves a lot of time for remote sensing image preprocessing. However, the remote sensing image data provided in GEE are limited, such as HJ-1A/B CCD, Gaofen, and VIIRS (Visible Infrared Imager/Radiometer Suite); other remote sensing image data are not used, so the applicability of the model to cross-sensor image data needs to be studied further. However, the band combination required by the NDVI is suitable for most satellite images, while the short-wave infrared band required by the FAI is not suitable for some satellite images. This problem can be solved by adding the VB-FAH algorithm [18] into the model so as to achieve fast and dynamic extraction of *U. prolifera* through multi-sensors.

(2) When there are more clouds in the image, the accuracy of the model will be affected. Although this situation can be avoided by choosing an image with a cloud cover less than 20%, the fact that *U. prolifera* distribution in the image with cloud cover of more than 20% is ignored. The cloud mask algorithm provided by GEE is not effective and will misclassify algae pixels into cloud pixels. Due to the failure of the FAI-COM, using NDVI-COM is a compromise method.

*4.2. The Critical Period for the Growth and Spread of U. Prolifera*

The growth of *U. prolifera* is inseparable from the suitable temperature and rich nutrients. As it happens, the special geographical conditions of Jiangsu shoal and the *Porphyra* culture environment provide support for the growth of *U. prolifera*. During the period from the end of May to the beginning of June, the early stage of *U. prolifera* was detected by remote sensing data, floating in the sea area near Jiangsu shoal, and the daily coverage increased day by day. Therefore, this period can be inferred as a critical period for the growth and spread of *U. prolifera* [63].

Figure 10 shows that the distribution of *U. prolifera* was originally discovered at the end of May in 2016 and 2019, covering areas of 108 km$^2$ and 3 km$^2$, respectively. The distribution of *U. prolifera* in early June of 2018, 2019, and 2020 was roughly similar, covering areas of 113 km$^2$, 194 km$^2$, and 159 km$^2$, respectively. The coverage of *U. prolifera* in early June of 2016 and 2017 was relatively high at 874 km$^2$ and 312 km$^2$, respectively. Overall, from the end of May to the beginning of June, a higher coverage area of *U. prolifera* distribution was observed during the five years. However, compared with 2016 and 2019, the overall scale of *U. prolifera* in 2017 and 2018 showed a decreasing trend (Figure 10). Wang found that, in 2017 and 2018, in the waters near Sheyang City, Jiangsu Province, early floating *U. prolifera* were salvaged and cleaned up. Therefore, the overall coverage of *U. prolifera* in these two years was relatively small [64].

Previous studies have shown that the growth rate of *U. prolifera* attached to the raft in April and May is not higher than 12.5% per day. When *U. prolifera* falls off the raft, the growth rate of *U. prolifera* floating on the sea will reach 20% per day [65]. The number of *U. prolifera* attached to the raft is very small, and difficult to detect by remote sensing satellite. Through UAV and field observation, Xing et al. learned that, in 2016, *P. yezoensis* facilities were recycled in mid-May [65]. While the recycling in 2017 and 2018 was at the end of May, the biomass of *U. prolifera* was 46% and 18% in the other two years of the same period;

follow-up studies showed that *P. yezoensis* facilities were recycled in early May in 2019, and the biomass of *U. prolifera* was higher in May. This delayed the recycling time of *P. yezoensis* facilities, and thus the delay in the shedding of attached algae into seawater may be the reason for the short time and low amount of *U. prolifera* growth and spread in 2017 and 2018. In summary, the early warning and monitoring of *U. prolifera* play a very important role. Early control and management of the source of *U. prolifera* (such as the recovery and cleaning of *P. yezoensis* culture facilities and the salvage of *U. prolifera*) can effectively curb the annual biomass.

## 5. Conclusions

Based on Google Earth Engine, this study used an adaptive threshold model to extract *U. prolifera*. The model was applied to Sentinel-2/MSI and Landsat8/OLI images to extract and analyze the distribution of *U. prolifera* in the South Yellow Sea, China from 2016 to 2020. The results show that:

(1) The model first performed Floating Algae Index (FAI) or Normalized Difference Vegetation Index (NDVI) algorithms on the preprocessed remote sensing images and then used the Canny Edge Filter and Otsu threshold segmentation algorithm to automatically extract the threshold.

(2) The model extraction of *U. prolifera* based on the FAI algorithm has higher accuracy ($R^2$ = 0.99, RMSE = 5.64) and better robustness. However, when the average cloud cover is more than 70% in the image (based on the statistical results of multi-year cloud cover information), the model based on the NDVI algorithm has better applicability and can extract the algae distributed at the edge of the cloud. Therefore, the final extraction results were generated by supplementing NDVI-COM results on the basis of FAI-COM extraction results in this paper. Further, the F1-score of *U. prolifera* extracted by the combined model was above 0.85.

(3) From 2016 to 2020, the interannual change in *U. prolifera* in the study area showed a decreasing trend. The overall outbreak scale of *U. prolifera* in 2017 and 2018 was relatively small, which was related to the delay in the recycle time of *P. yezoensis* culture facilities in the Northern Jiangsu shoal and the artificial salvage of *U. prolifera* in May. In contrast, early warning and cleanup measures were not taken in 2019, and the outbreak scale of *U. prolifera* rebounded.

Compared with the traditional threshold selection and the Otsu threshold algorithm, the model proposed in this paper is more convenient and accurate for the extraction of *U. prolifera*, and it has high robustness to environmental changes. This article focused on a case study in the South Yellow Sea, China, where green tides frequently occur. It is hoped that it can bring scientific and technical support to the monitoring and early warning of green tides in the study area.

**Author Contributions:** Conceptualization, G.Z., G.X.; methodology, G.Z.; software, H.L.; validation, G.Z., J.W.; formal analysis, L.N.; investigation, Y.H.; resources, G.Z.; data curation, G.Z.; writing—original draft preparation, G.Z.; writing—review and editing, M.W.; visualization, G.X.; supervision, G.X.; project administration, G.X.; funding acquisition, G.X. All authors have read and agreed to the published version of the manuscript.

**Funding:** This work is supported by the Shenzhen Science and Technology Program (Grant No. KQTD20180410161218820), the National Natural Science Foundation of China (No. 42071385), and the Shandong Natural Science Foundation (No. ZR2019MD041).

**Institutional Review Board Statement:** Not applicable.

**Informed Consent Statement:** Informed consent was obtained from all subjects involved in the study.

**Data Availability Statement:** The data presented in this study are available upon request from the corresponding author.

**Acknowledgments:** The remote sensing data of Google Earth Engine are available online at https://earthengine.google.com/ (accessed on 10 July 2021). Data of the *U.prolifera* by the Ministry of Natural Resources North China Sea Administration are available online at http://ncs.mnr.gov.cn/ (accessed on 10 July 2021). The statistical data of *U.prolofera* by the China Marine Disasters Bulletin from 2008 to 2019 are available online at http://www.mnr.gov.cn/ (accessed on 10 July 2021). In the end, the authors like to thank the anonymous reviewers for their efforts and constructive comments to improve the quality of this paper.

**Conflicts of Interest:** The authors declare no conflict of interest.

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
