# Peer review of "Adaptive Threshold Model in Google Earth Engine: A Case Study of Ulva prolifera Extraction in the South Yellow Sea, China"

_remotesensing, doi:10.3390/rs13163240_

Round 1
Reviewer 1 Report
Review comments for “Adaptive Threshold Model in Google Earth Engine: A Case Study of Ulva prolifera Extraction in the South Yellow Sea, China”
Major review
Authors did not show truth data to validate. Although Xing et al. (2019), cited in this manuscript, showed increasing trend from 2007 to 2018, authors said the decreasing trend during 2016-2020 is similar with it. Showing decreasing trend for 5 years seems not proper to discuss the tendency of Ulva distribution even without the confidence of the new method. I suggest 1) comparing the detected pixels’ spectral reflectance with in-situ green tide’s spectral reflectance, 2) comparing MODIS result for 20 years with Xing et al. (2019) or the China Marine Disasters Bulletin, and 3) comparing Landsat-8 + Sentinel-2 to MODIS result.
[Abstract]
- Can NDVI be calculated with any kind of cloud? There is not a proper reference for that in the manuscript.
[Introduction]
- Reference for L45-49 is missing.
- Second paragraph of Introduction is too narrative and hard to get the idea for understanding this study. It can be reduced concisely.
[Data]
- What kind of atmospheric correction was used for GEE Landsat 8 and Sentinel-2?
- FAI was invented for Rayleigh corrected reflectance (Rrc) according to Hu (2009). Is GEE product that authors used for this study Rrc?
- What are the ? and ?ℎ parameters?
- Keep same capitalization of the Canny Edge Filter in the text.
[Results]
- L249-251 should be re-written considering the relative expression.
- L254-260 is not a result of this study. Summarizing institution observation or previous studies about green tide in the Southern Yellow Sea can be a result.
- Fig. 4d, 4j, 6e seem like as hyperspectral data not multispectral reflectance of Sentinel-2. More important thing is matching green tide’s spectral characteristic with the detected pixels’ in Sentinel-2 & Landsat-8.
- Fig. 5: lack of information in the caption.
- Fig. 8 and 10: 2016 and 2019 are higher on both satellites, but rest of the years’ coverage area values are somewhat different.
- L438: while that of 2016 and ‘2019’ is relatively large.
- It seems there is no need to mention author’s future goal in the manuscript.
Reviewer 2 Report
Many thanks to the authors for the revision of the manuscript. The manuscript was improved a lot. However, I still have the following concerns:
- The authors’ should be clear in the text why these high-resolution data (L8 and S2) were used in HABs monitoring in this study instead of medium-resolution data. It is insufficient to uses a linear trend to show the representation of high-resolution data. The daily /monthly mean /annual mean should be compared or a detailed analysis like Cao2019 should be performed. Moreover, the authors still do not address/explain the problem that the S2/L8 data did not cover the whole study area. These disadvantages of high-resolution data also should be mentioned in the text.
- Validation using visually interpreted ground truth is recommended to be performed before cross-validation.
- Fig. 13 is insufficient to support the advantages of FAI/NDVI-COM compared to the fixed FAI/NDVI threshold. I think it is better to provide the zoom of some details may.
Reviewer 3 Report
This manuscript used GEE and Landsat-8/Sentinel-2 to extract the U. prolifera in the Yellow sea from 2016 to 2020. This topic is not novel. As reviewed in the introduction, there are a number of studies to extract U. prolifera here using various sensors. Although this study provided an automatic extraction with GEE, some issues could still be address. First, this manuscript should clarify why you use S2 and L8 to monitor U. prolifera. Although they have a fine spatial resolution, the 250m of MODIS should be enough for the coastal area. Second, the methodology is not very clear; the readers would like to know every step, in particular the overall flow. Thirdly, the study generated a five-year dataset of U. prolifera; however, L8 and S2 had a long revisit time, the derived trend may be affected by the image number.
- Line 16: U. prolifera should be defined in line 14.
- Line 25: Why does this model include all advantages of these methods?
- The abstract was in a mess, suggesting to improving it following objective, method, result, and conclusion. In particular, the current texts about the method did not provide a clear understanding.
- Line 27: This is not your contribution. Suggesting to remove it.
- The cloud flag in reflectance product possibly recognized the floating U.prolifera as clouds.
- Line 179-180: Please provide references.
- Figure 2 (a)and (b) was not clear. What is the gray line in Figure 2c? Please improve them.
- Line 243 and 244: Confused. Entire study area or one OLI/MSI image? Why selected 70%?
- In the method part, the manuscript should describe how to generate the final U. prolifera results using FAI and NDVI. Here, the readers only learn that you developed two models, NDVI-DOM and FAI-DOM. Two models would generate two results.
- Line266-267: How to visual interpretation? Details should be mentioned.
- Figure 4:panels (d) and (j) should be plotted at a line rather than a curve. And this plot should use same color for each class point (water and U. prolifera).
- Lines 332 and 343: It is not clear what the purpose of this paragraph. Furthermore, I am still confused why NDVI could extract U. Prolifera in the cloudy image. Please add some graphs to illustrate it.
- Figure 7: Relative to the area comparison, the readers, would like to see the result of pixel by pixel.
- Section 3.2: This manuscript generated the U.prolifera using MSI and OLI. Did you compare the results with MODIS? MSI and OLI had a long revisit time; however, the U. prolifera had a fast variation. Are they enough to monitoring spatial and temporal distribution? Suggesing to introduce how many satellite images were used in the time series each month since 2016.
- Discussion: 4.1.1 These are not real advantages of your model. Automatic extraction and high spatial mapping may be more suitable. Also, the uncertainty of the model should be further addressed. What factors would affect the extraction accuracy?
Round 2
Reviewer 1 Report
- I recommend the authors to consider removing the regression line in figures or showing 20 years of MODIS result.
- About the Rrc, it is still unclear authors used Rrc or just GEE reflectance product.
- Fig 7. ROI and full name should be switched.
Author Response
Dear Reviewers,
Thanks again for your comments concerning our manuscript. Please see the attachment.

Reviewer 3 Report
Thanks for the revision. The current version has addressed my concerns basically. Suggesting authors checking and editing the spellings and format throughout manuscript before publication.
Author Response
Dear Reviewers,Thanks for your comments concerning our manuscript, we have checked
and edited the spelling and format of the whole manuscript before
publication. All revisions made to the manuscript were marked up
using the “Track Changes” function.Once again, thank you very much for your comments and suggestions.
This manuscript is a resubmission of an earlier submission. The following is a list of the peer review reports and author responses from that submission.
Round 1
Reviewer 1 Report
Review comments on “Adaptive Threshold Model in GEE: A Case Study of Ulva prolifera Extraction in the South Yellow Sea, China”
This manuscript shows new methodology for Ulva detection especially using new and fast Google earth engine. However, there are too many examples, which can be concisely reduced. There are many broken sentences, abbreviations without full name in the Abstract and Introduction, which makes hard to keep reading the manuscript smoothly and decreases the expectation of this study result. Most of all, it is recommended to compare the result with ground truth or previously proved result for validation.
- Title: GEE ->Google Earth Engine
- Abstract: broken sentences, what is P. yezoensis?
- Keep same name for Landsat-8/OLI in the manuscript
Introduction
- Ulva prolifera can be mentioned U. prolifera after one full name.
- Still there are broken sentences.
Introduction
- What is P. yezoensis’ full name or common name?
- What is the harmful effect of Ulva?
- L60: Does Ulva sank and decomposed cause harmful effect like hypoxia?
- L64: spectral wave->spectral reflectance or spectral characteristics
- L69: Reference 16 does not mention ulva or green algae. The reference 16 and 17 are not enough refer all the mentioned satellite.
- L75: What does the best value of algae mean?
- L80: Reference needed or it should be studied in this study.
- L85: This sentence is too long and can be divided in two different sentences. The meaning is confusing. Does it mean that many scholars have realized the fast extraction and monitoring of Ulva can be possible by AI application for satellite data preprocessing?
- L90: Full name of GOCI is missing.
- L111: What is COM (FAI-COM, NDVI-COM)?
- L141: Sentinel-2’s revisit frequency is 5 days in the most of region.
Methods
- L148-154: Check the instruction removed.
- Fig 2: What do the lines in Fig 2c represent?
Result
- L243: I do not agree. Validation with ground truth data (from cruise or previous study) is needed or previously proved remote sensing result. Comparing two different unproved result represents the characteristics of the satellites’ sensors. In this manner, what makes the different result must be addressed comparing each satellites’ spectral reflectance at Ulva pixels.
- Fig 4-7: What represent A&D’s color difference?
- Fig. 4-12: These examples can be reduced as one or two. Same date, two methods, two satellites.
In the method(L130-131), Landsat-8 is the main and Sentinel-2 is for validation. But the result shows Sentinel first.
- Fig 14: How about showing year-to-month/date of the extent? To show that seasonal variation? But I wondered if the coarse data acquisition of two satellites was proper/enough to analyze the trend. If this method can be applicable to GOCI or MODIS, these satellites data is more suitable for temporal variation analysis.
- Fig 14: Was it calculated in the same research area considering cloud cover? Is it from both satellites?
- Fig 18: A’s explanation is missing. Looking at C, E, FAI and Otsu look not good to be used for the Ulva detection. If authors want to show the limitation of these previous methods, it would be better to show before your main result to emphasize the need for new methodology.
- L600: There are different kinds of harmful algal bloom by color. Specifically mention green tide or Ulva is better.
Author Response
Dear Reviewers:
Thank you very much for your comments concerning our manuscript entitled “Adaptive Threshold Model in Google Earth Engine: A Case Study of Ulva prolifera Extraction in the South Yellow Sea, China (remotesensing-1262225)”. Those comments are all valuable and very helpful for revising and improving our paper, as well as the important guiding significance to our research. We have made a careful revision on the original manuscript. All revised portions are marked in red in the revised manuscript which we would like to submit for your kind consideration.
If you have any further question about this paper, please don’t hesitate to contact us. Many thanks.
Sincerely yours,
Guangzong Zhang

Reviewer 2 Report
Comments for “Adaptive Threshold Model in GEE: A Case Study of Ulva prolifera Extraction in 2 the South Yellow Sea, China (remotesensing-1262225)”
Based on the GEE platform, this study aimed to combine the OSTU and Canny Edeg Filter algorithms to generate the distribution map of Ulva prolifera in the South Yellow Sea of China from 2016 to 2020, analyze the spatial and temporal changes of Ulva prolifera in the five years, and discuss the reasons for the changes. The manuscript provided a large number of images and results to prove the accuracy of the developed algorithms. However, it still needs to be improved considerately. Some major comments are shown as below.
- The background is not fully introduced, and the work done by Xing et al. is not introduced.
- The developed method of this paper needs to be innovated. NDVI, FAI and OTSU are all commonly used methods and/or indicators. In this paper, the Canny Edge Filter of NDVI and FAI was firstly carried out, and then the OTSU segmentation was carried out. This is a very simple combination, lacking innovation, and the purpose of edge detection is not very clear.
- The application results of the developed method need to be better verified. This paper only lists some monitoring results of typical dates to illustrate the application effect of the method, which is redundant and many of them are unnecessary. Instead, I want to see statistical results, as shown in Figure 13, comparing the results for different dates. However, Figure 13 is just a comparison of the results on a few dates, and it would have been better to add more results on different dates.
- Although this paper focused on the time range of 2016-2020, as shown in Figure 14, only Ulva prolifera on some dates were extracted. Monitoring results on only five days were obtained in 2017, 2018 and 2020. In this case, the results cannot reflect the change trend of Ulva prolifera from 2016 to 2020.
- The reasons discussed in Section 4.2 for the change of Ulva prolifera during 2016-2020 are lack of data support and are not convincing.
- The density of Ulva prolifera was not considered in the extraction.
- In Figure 3, why Edge Filter and OTSU segmentation were done two times ?
- There are a large number of grammatical and expressive errors in the English writing, and even the writing instructions (Lines 148-153) are retained, which indicate that the author did not make enough checks before submitting the manuscript.
Author Response

(The authors gave the same response as above.)

Reviewer 3 Report
The paper entitled "Adaptive Threshold Model in GEE: A Case Study of Ulva prolifera Extraction in the South Yellow Sea, China" provides an adaptive threshold model to extract Ulva prolifera. However,
I have some significant concerns about this manuscript:
- The manuscript mentioned that many previous studies had used medium-resolution sensors such as MODIS to extract Ulva prolifera, while this study used high-resolution data (i.e., S2-MSI and L8-OLI). Therefore, the authors need to clarify the reasons/advantages of high-resolution data used instead of medium-resolution sensors in this study. Of course, this is a good attempt to use multi-source high-resolution data in HABs monitoring. However, both MSI and OLI are tens of meters images, and the MSI results are also not strictly quantitatively validated to be regarded as the ground truth values for OLI. Therefore, it is recommended to use ROI points with visual inspection or meter/sub-meter level images as ground truth and use S2-MSI and L8-OLI to monitor HABs jointly.
- I noticed some problems that need to be addressed for the long-time series data products: how to ensure the consistency of long-term series results when the adaptive threshold was used instead of a fixed unified threshold; L8-OLI data in a single path/date cannot completely cover the whole study area; HABs are highly variable, but the time resolution of OLI data is low; the impact of cloud coverage.
- Finally, GEE is a good tool for processing large amounts of data. But for now, the manuscript is only a case study. Therefore, it is recommended to add a prospect/implication of a more considerable amount of remote sensing image processing to highlight the advantages of GEE.
Specific comments
- Line 137: The TOA data were used in this study, so how sensitive is the model proposed to the atmosphere?
- Figure 3: How to mask land?
- Line 148-154: Delete
- Line 187-188: Hu et al. (2010) has already proposed an automated threshold extraction method, but a time‐independent threshold value was obtained using the statistical method from FAI thresholds in all individual images
- Will OTSU always get a threshold even if no HABs occurred?
- It isn't easy to see the sub-graph of the spectrum curve clearly in Figure 4 and similar figures. Is this the satellite spectrum or the in situ measured spectrum?
- Figures 4-7, as well as corresponding descriptions, can be merged and summarized. Show the map of FAI and NDVI.
- Figures 9-12 are the same as above
- Line 516-517: How to understand the model containing two indices?
- Line 522-536: It is not fair to use -0.004 and 0 to compare here. The fixed unified threshold should be obtained first with the method proposed by Hu et al. (2010) or a similar method.
- Figure 18: E-F was obtained from FAI or NDVI; show the map of FAI and NDVI; finally, I prefer that all figure captions should be clear enough to understand independently.
- Line 540-542: These are not the limits of the model.
- 2. Analysis on the area change of Ulva prolifera in five years: Are there any discoveries in addition to similar results to existing studies?
- Explain why the model is not sensitive to cloud and sunglint.
- Hu, C., Lee, Z., Ma, R., Yu, K., Li, D., Shang, S., 2010. Moderate Resolution Imaging Spectroradiometer (MODIS) Observations of Cyanobacteria Blooms in Taihu Lake, China. Journal of Geophysical Research, 115
Author Response

(The authors gave the same response as above.)

Round 2
Reviewer 1 Report
2nd comments for “Adaptive Threshold Model in Google Earth Engine: A Case Study of Ulva prolifera Extraction in the South Yellow Sea, China”
The authors tried to improve the manuscript more concisely with explicit language. However, there are two significantly unignorable concerns.
- Ground truth validation.
-Authors are insisting comparing two unproved result is validation, but I sincerely do not agree with it again. Because authors’ method select data for training (or other process) from multiple different method. I recommend authors to survey previous ground truth data and add section for the validation with ground truth data. If there is no such in situ data, then authors must change the term validation to comparison.
- Temporal trend by time series result.
In the introduction, the authors mentioned that increasing Ulva is the problem. I already asked if the amount of data from Landsat and Sentinel was enough to show a temporal trend or seasonal variation. Although authors replied that they considered it enough, Fig. 8 & 10 do not support the increasing trend. If the authors show a trend in this study, GOCI and MODIS are more appropriate considering the channels and shorter data intervals.
Reviewer 2 Report
Many thanks to the author for his revision of the manuscript.
I fully understand the difficulty of revising the manuscript, but I don't think the author has substantially modified the manuscript according to the comments of the first-round review. The manuscript still needs further improvement.
Although the conclusion in this paper is correct, it still needs to be illustrated by MODIS monitoring results.
Reviewer 3 Report
Many thanks to the authors for the revision of the manuscript. The manuscript was slightly improved compared to its previous versions. However, the authors have not modified the manuscript substantially, and I am not satisfied with the authors’ reply to comments of the first-round review, especially the followings:
- Considering the small width of OLI and MSI data that cannot cover the entire study area and the annual images with no more than 15 scenes, how did the author draw a conclusion that the results could represent the temporal and spatial distribution of HABs in the study area? The author emphasizes that the results have been compared with published data, so please show the comparison and give some statistically significant indices. And it is recommended to compare with the trend of MODIS results (including daily /monthly mean /annual mean) to illustrate the accuracy of the temporal and spatial distribution of OLI and MSI.
- The FAI/NDVI unified thresholds used for comparison need to be re-calibrated with the method by Hu2010. It should be noted that Hu2010 used Rayleigh-corrected reflectance, while the authors used the reflectance product (please clarify it is the Atmospherically corrected surface reflectance or the top-of-atmosphere (TOA) reflectance in the data and method). And -0.004 was obtained for cyanobacteria blooms in Lake Taihu, as the authors replied in another comment "The applicability of the study area (Southern Yellow Sea, China) needs further study.".
- The current model that the authors used was FAI-COM or NDVI-COM separately, and no model was used with the combinations of the two indices. Therefore, it is inaccurate to emphasize that the model is insensitive to sunglint/cloud. It is recommended to say the FAI-COM is insensitive to sunglint, while the NDVI-COM is not sensitive to the cloud.
- Other replies also need to be considered again.